# An engineered monomer binding-protein for α-synuclein efficiently inhibits the proliferation of amyloid fibrils

Emil Dandanell Agerschou[1†], Patrick Flagmeier[2,3†], Theodora Saridaki[4], Céline Galvagnion[5,6], Daniel Komnig[4], Laetitia Heid[1], Vibha Prasad[4], Hamed Shaykhalishahi[1], Dieter Willbold[1,7], Christopher M Dobson[2,3], Aaron Voigt[4], Bjoern Falkenburger[4,8,9]*, Wolfgang Hoyer[1,7]*, Alexander K Buell[1,10]*

[1]Institut für Physikalische Biologie, Heinrich Heine University Düsseldorf, Düsseldorf, Germany; [2]Department of Chemistry, University of Cambridge, Cambridge, United Kingdom; [3]Centre for Misfolding Diseases, University of Cambridge, Cambridge, United Kingdom; [4]Department of Neurology, RWTH Aachen University, Aachen, Germany; [5]RG Mechanisms of Neuroprotection, German Centre for Neurodegenerative Diseases (DZNE), Bonn, Germany; [6]Department of Pharmacology and Drug Design, University of Copenhagen, Copenhagen, Denmark; [7]Institute of Complex Systems (ICS-6), Structural Biochemistry, Forschungszentrum Jülich, Jülich, Germany; [8]Department of Neurology, Dresden University Medical Center, Dresden, Germany; [9]JARA BRAIN Institute II, Julich and Aachen, Germany; [10]Department of Biotechnology and Biomedicine, Technical University of Denmark, Kgs. Lyngby, Denmark

*For correspondence:
bjoern.falkenburger@ukdd.de (BF);
wolfgang.hoyer@uni-duesseldorf.de (WH);
alebu@dtu.dk (AKB)

†These authors contributed equally to this work

Competing interests: The authors declare that no competing interests exist.

**Abstract** Removing or preventing the formation of α-synuclein aggregates is a plausible strategy against Parkinson's disease. To this end, we have engineered the β-wrapin AS69 to bind monomeric α-synuclein with high affinity. In cultured cells, AS69 reduced the self-interaction of α-synuclein and formation of visible α-synuclein aggregates. In flies, AS69 reduced α-synuclein aggregates and the locomotor deficit resulting from α-synuclein expression in neuronal cells. In biophysical experiments in vitro, AS69 highly sub-stoichiometrically inhibited both primary and autocatalytic secondary nucleation processes, even in the presence of a large excess of monomer. We present evidence that the AS69-α-synuclein complex, rather than the free AS69, is the inhibitory species responsible for sub-stoichiometric inhibition of secondary nucleation. These results represent a new paradigm that high affinity monomer binders can lead to strongly sub-stoichiometric inhibition of nucleation processes.

## Introduction

Cytoplasmic aggregates of the protein α-synuclein are the pathological hallmark of Parkinson's disease (PD) and other synucleinopathies (*Spillantini et al., 1997*). Point mutations in the α-synuclein gene or triplication of the α-synuclein locus are associated with familial forms of PD, and the α-synuclein locus is a genetic risk factor for sporadic PD (*Obeso et al., 2017*). α-synuclein aggregate pathology was demonstrated to propagate from neuron to neuron (*Desplats et al., 2009*), and recent work has focused on understanding the cellular and molecular events in this process. From a therapeutic perspective, α-synuclein aggregation is thought to be the underlying cause of PD and remains the focus of causal therapeutic strategies. The link between α-synuclein

aggregation and PD has been known for two decades (*Spillantini et al., 1997*; *Conway et al., 1998*); however, translation of this scientific discovery into a therapy has proven challenging. From the first description of small molecules that inhibit α-synuclein aggregation in 2006 (*Masuda et al., 2006*), the search for promising compounds has continued (*Wagner et al., 2013*; *Tóth et al., 2014*; *Wrasidlo et al., 2016*; *Perni et al., 2017*; *Kurnik et al., 2018*). While the first small molecules also inhibited the aggregation of tau and amyloid-β, more recent compounds bind α-synuclein more selectively and show reduced α-synuclein toxicity in mouse models of PD (*Wrasidlo et al., 2016*). We have taken a different strategy by engineering a protein, the β-wrapin AS69, to induce formation of a β-hairpin in monomeric α-synuclein upon binding (*Figure 1a*) (*Mirecka et al., 2014*). AS69 was selected by phage display (*Mirecka et al., 2014*) from protein libraries based on ZAβ3, an affibody against the amyloid-β peptide (*Hoyer et al., 2008*; *Hoyer and Härd, 2008*; *Luheshi et al., 2010*). AS69 thus not only binds α-synuclein with high and approximately constant affinity throughout the pH range most relevant for α-synuclein aggregation (*Buell et al., 2014a*; *Figure 1b,c*), but also induces a specific conformational change - akin to molecular chaperones (*Muchowski and Wacker, 2005*).

AS69 induces local folding of the region comprising residues 37–54 into a β-hairpin conformation in the otherwise intrinsically disordered, monomeric α-synuclein (*Figure 1a*). The critical role of this region for α-synuclein aggregation is indicated by the cluster of disease-related mutation sites (*Figure 1a*). Accordingly, modification of the local conformation by, for example, introduction of a disulfide bond strongly modulates aggregation (*Shaykhalishahi et al., 2015*). Sequestration of residues 37–54 of monomeric α-synuclein by AS69 inhibits the amyloid fibril formation of α-synuclein under conditions of vigorous shaking of the solution even at highly sub-stoichiometric ratios (*Mirecka et al., 2014*). Amyloid fibril formation, however, is not a one-step process but can be decomposed into different individual steps, including primary and secondary nucleation and fibril elongation. With vigorous shaking, for instance, primary nucleation can occur readily at the air-water interface (*Campioni et al., 2014*) and fibril fragmentation induced by the shaking amplifies the number of growth-competent fibril ends (*Xue et al., 2009*). To validate AS69 as a potential therapeutic

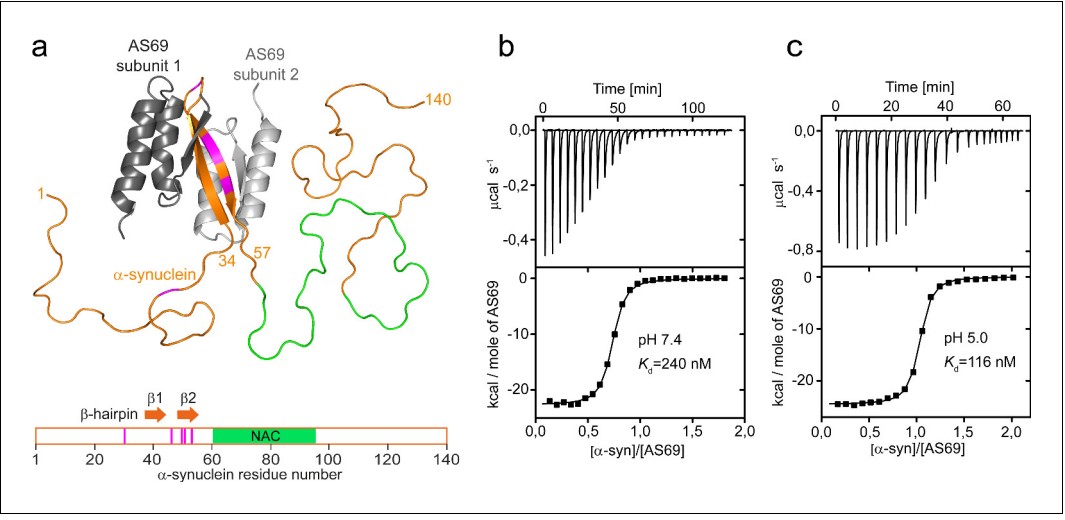

**Figure 1.** AS69 binds to monomeric α-synuclein, inducing local folding of the region comprising residues 37–54 into a β-hairpin conformation. (a) Structural model of the AS69:α-synuclein complex based on NMR (pdb entry 4BXL) (*Mirecka et al., 2014*), generated with PyMOL (The PyMOL Molecular Graphics System, 1.2; Schrödinger, LLC.). AS69 (grey) is a disulfide-linked homodimer. α-Synuclein (orange) locally adopts β-hairpin conformation, while the remainder of the molecule, including the hydrophobic NAC segment (green), remains intrinsically disordered (*Mirecka et al., 2014*). Positions at which disease-related mutations have been identified are given in magenta. (b,c) The affinity of AS69 to α-synuclein at pH 7.4 (b) and pH 5.0 (c) analyzed by isothermal titration calorimetry (ITC) experiments. Titration of 420 µM α-synuclein into 47 µM AS69 in 20 mM sodium phosphate, 50 mM NaCl, pH 7.4 (b), or 320 µM α-synuclein into 32 µM AS69 in 20 mM sodium acetate, pH 5.0 (c), at 30 °C. The upper panels show the baseline-corrected instrumental response. The lower panels show the integrated data (filled squares) and the fit to a 1:1 binding model (continuous line).

agent, we therefore tested its biological effects in cellular and animal models, and found it to be a highly efficient inhibitor of $\alpha$-synuclein aggregation and associated toxicity. In addition, we designed a set of experimental conditions to measure selectively the effect of AS69 on specific steps of $\alpha$-synuclein aggregation. We found that AS69 is able to efficiently interfere with both the lipid-induced formation and the auto-catalytic amplification of $\alpha$-synuclein amyloid fibril formation. These inhibitory effects on nucleation are observed even in the presence of a large excess of $\alpha$-synuclein monomer, which is expected to sequester AS69 into inhibitor-monomer complexes. We show evidence that the secondary nucleation of $\alpha$-synuclein can be inhibited by the $\alpha$-synuclein-AS69 complex and, therefore the inhibitory effect of AS69 on this crucial step of aggregate amplification is unaffected by even large excess concentrations of free $\alpha$-synuclein monomer.

## Results

### Co-expression of AS69 reduces visible $\alpha$-synuclein aggregates in cell culture

First, we explored the effect of the expression of AS69 on the viability of living cells and the association of $\alpha$-synuclein in a cellular environment. In these model systems we not only expressed WT $\alpha$-synuclein but also the A53T variant, which has been associated with familial PD and which produces aggregates more quickly than the WT protein (*Conway et al., 1998*; *Flagmeier et al., 2016*). We first used bimolecular fluorescence complementation (BiFC) to probe whether AS69 can interfere with formation of oligomeric $\alpha$-synuclein species in living HEK293T cells (*Falkenburger et al., 2016*). Constructs of WT and A53T $\alpha$-synuclein were tagged with the C-terminal segment of the fluorescent protein Venus (synuclein-VC) or with the complementary N-terminal segment of this protein (VN-synuclein) (*Figure 2a*). Neither of the two Venus fragments shows significant fluorescence by itself, but together they can generate a functional fluorescent protein (*Bae et al., 2014*) and hence function as a reporter for protein-protein interaction. We then transfected HEK293T cells with both synuclein-VC and VN-synuclein, in addition to AS69 (or LacZ as a control) and determined by flow cytometry the fraction of cells that displayed Venus fluorescence (*Figure 2b*, the raw data can be found in the table in *Figure 2—source data 1*). In the absence of AS69, the fraction of fluorescent cells was larger with the expression of A53T-$\alpha$-synuclein than WT-$\alpha$-synuclein (*Figure 2b*, p<0.05, two-way ANOVA). Co-expression of AS69 with both variants reduced the number and fraction of fluorescent cells (*Figure 2b*, p<0.05 for WT and p<0.01 for A53T, two-way ANOVA). AS69 did not, however, significantly affect the total quantity of $\alpha$-synuclein in the cells, as determined from immunoblots (*Figure 2c and d*). This finding is consistent with the hypothesis that the effects of AS69 in this cellular model system result from inhibition of a direct interaction between $\alpha$-synuclein molecules, and not from an enhanced clearance of $\alpha$-synuclein. Despite the enhanced affinity for self-interaction which the fluorescence complementation tag might convey to $\alpha$-synuclein compared to the untagged protein, the affinity for AS69 is high enough to sequester a significant proportion of the $\alpha$-synuclein in living cells.

Having established that $\alpha$-synuclein and AS69 can interact in cells, we next probed the effects of AS69 on the formation of larger, optically visible aggregates of $\alpha$-synuclein by transfecting HEK293T cells with A53T-$\alpha$-synuclein tagged with enhanced green fluorescent protein (EGFP) as previously described (*Opazo et al., 2008*; *Karpinar et al., 2009*; *Dinter et al., 2016*; *Figure 2e*). The distribution of EGFP within transfected cells was classified as 'homogenous', 'containing particles' or 'unhealthy' (rounded cells that in time-lapse microscopy were observed to subsequently undergo apoptosis). Co-expression of AS69 with A53T $\alpha$-synuclein led to an increase in the fraction of cells with a 'homogenous' distribution of EGFP and fewer cells showed $\alpha$-synuclein particles relative to those cells without AS69 (*Figure 2f*). These findings indicate that the co-expression of AS69 reduces formation of visible aggregates in cultured human cells.

### Co-expression of AS69 rescues A53T $\alpha$-synuclein-dependent phenotype in *Drosophila melanogaster*

Subsequently, we tested the effects AS69 has in *Drosophila melanogaster* (fruit flies) expressing untagged A53T-$\alpha$-synuclein in neurons (*Figure 3*). In the absence of AS69, these flies show a progressive reduction in the spontaneous climbing (i.e. neuronal impairment) between 15 and 25 days

of age (*Butler et al., 2012*; *Dinter et al., 2016*; illustrated in *Figure 3a*). We then generated flies co-expressing either AS69 or GFP (as a control) with A53T $\alpha$-synuclein in neurons. Flies expressing

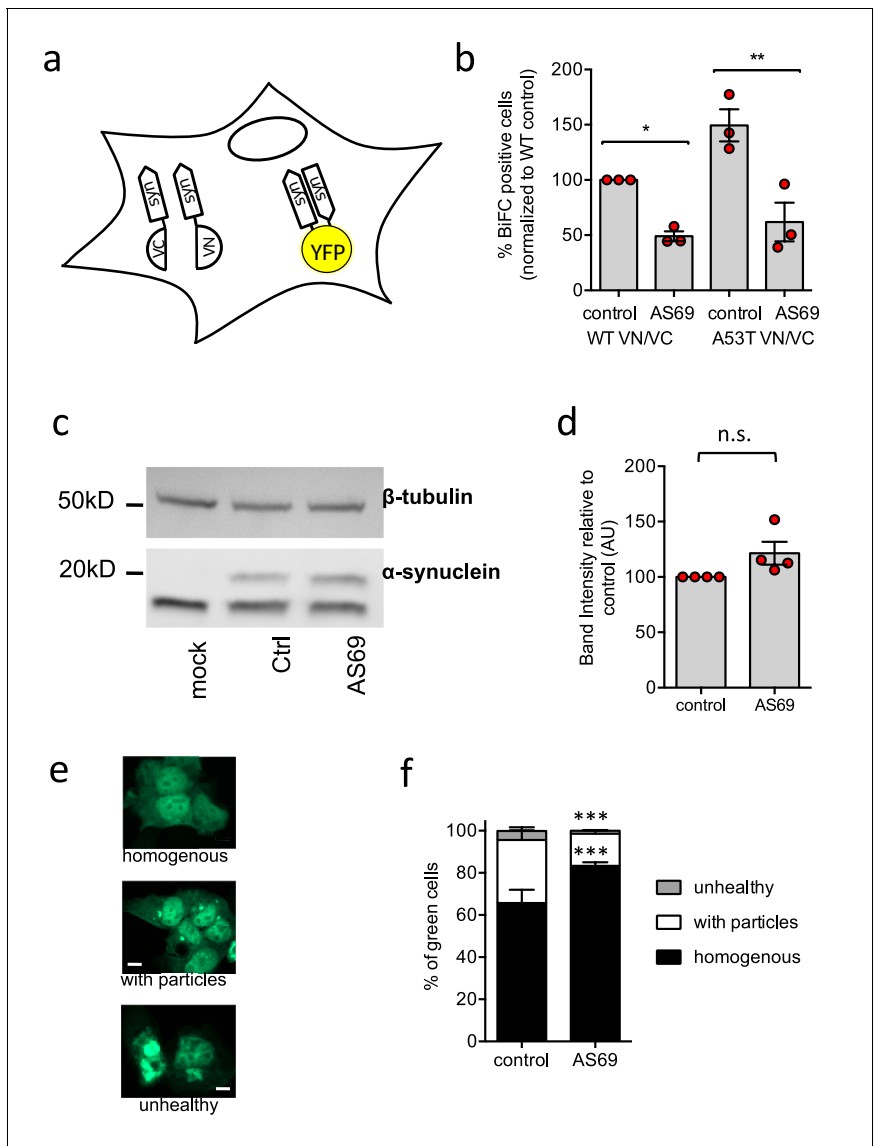

**Figure 2.** AS69 reduces aggregation of $\alpha$-synuclein in cellular models. (**a**) Schematic representation of bimolecular fluorescence complementation where $\alpha$-synuclein is tagged by either the C-terminal (VC) or the N-terminal (VN) fragment of the Venus protein. In dimers or larger oligomers of $\alpha$-synuclein, the two Venus fragments can form a functional fluorescent protein. (**b**) The percentage of cells with BiFC fluorescence as determined by flow cytometry. HEK293T cells were transfected with $\alpha$-synuclein (WT or A53T), fused to the VN or VC fragment and either LacZ (control) or AS69. Displayed are the results of n = 3 independent experiments and mean ± SEM. In each experiment, 75,000 cells were analyzed per group. Results were compared by one-way ANOVA, results of Sidak's posthoc test depicted. (**c**) Immunoblot of lysates of cells transfected with EGFP-tagged $\alpha$-synuclein and, in addition, AS69 or LacZ (control), developed with antibodies against $\alpha$-synuclein (band just below 20 kDa, note that only the upper band reports $\alpha$-synuclein, *Dinter et al., 2016*) and $\beta$-tubulin (band just below 50 kDa), the latter as a loading control. (**d**) Quantification of n = 4 independent blots as described in (**c**). Results were compared by t-test. (**e**) HEK293T cells were transfected with EGFP-tagged $\alpha$-synuclein and the distribution of fluorescence was classified into the depicted groups. (**f**) Summarized results of n = 3 independent experiments with n = 300 cells classified per group in each experiment (mean ± SEM). Results were compared by two-way ANOVA and Sidak's posthoc test.

The online version of this article includes the following source data and figure supplement(s) for figure 2:

**Source data 1.** Raw cell counts of cells from the three independent experiments shown in *Figure 2b*.

**Figure supplement 1.** Complete Western blot (*Figure 2c*) from cell culture lysates showing the loading control with $\beta$-tubulin at 50 kD, two nonspecific bands visible also in mock transfected cells, that is without $\alpha$-synuclein expression, and one specific band just below 20 kD (*).

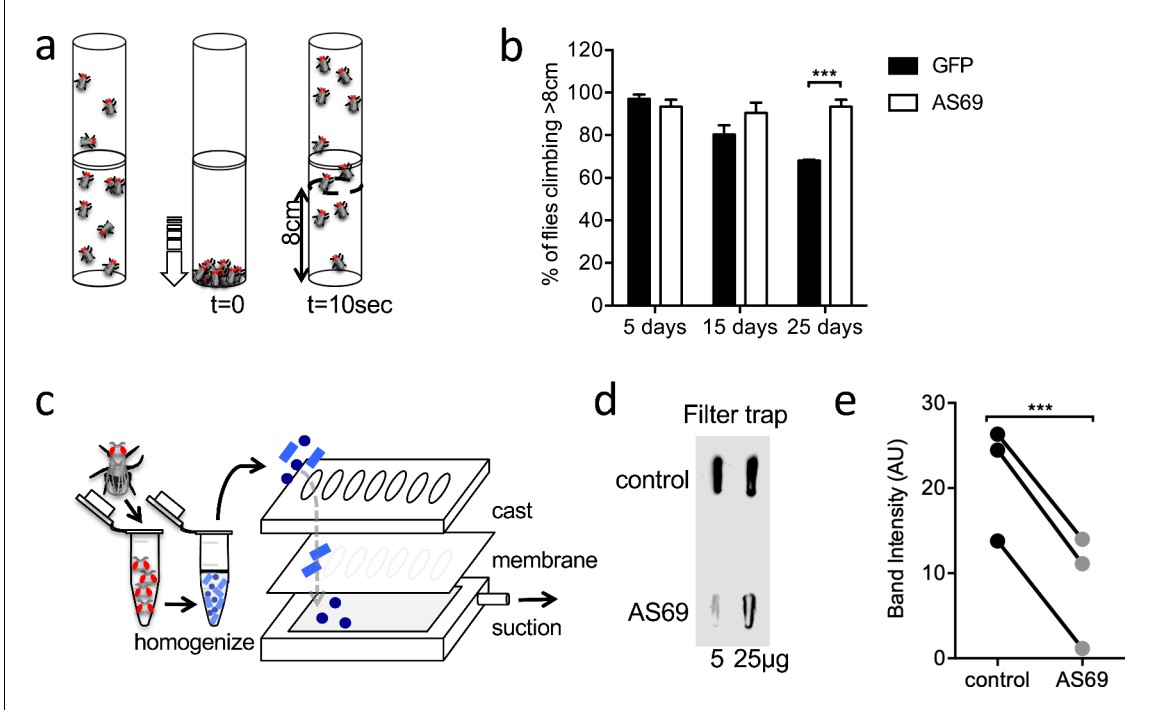

**Figure 3.** AS69 rescues the motor phenotype and reduces α-synuclein aggregation in *Drosophila melanogaster*. (a) Schematic representation of the climbing assay. The vials are tapped to move the flies to the base of the vial, and thereafter the flies climb towards the top of the vial; in this experiment the number of flies climbing 8 cm in 10 s was determined. (b) Performance in the climbing assay of *Drosophila melanogaster* expressing A53T-α-synuclein and either AS69 or GFP in neurons. At each time point, n = 30 flies were assayed per genotype; similar findings were observed for eight different lines expressing AS69. Results were compared by two-way ANOVA and Sidak's posthoc test. (c) Schematic representation of the filter trap assay in which aggregates in the protein lysate are retained by a membrane, which is subsequently developed in the same manner as an immunoblot. (d) Results of the filter trap assay from lysates of control flies and flies expressing AS69 in addition to A53T-α-synuclein in all neurons. Two different quantities of the protein lysate were applied in each case, 5 and 25 µg. (e) Summary of the quantification of n = 3 dot blots as in (d). Only the 25 µg band was quantified. Results were compared by t-test.

AS69 and A53T α-synuclein showed preserved climbing behaviour (*Figure 3b*, two-way ANOVA), demonstrating that neuronal expression of AS69 reduces the phenotype in this fly model of A53T α-synuclein toxicity. We further went on to determine whether or not the observed effect of AS69 on climbing behaviour could result from a reduction in the number of α-synuclein aggregates and used flies expressing in all neurons one copy of A53T-α-synuclein fused to VC, one copy of A53T-α-synuclein fused VN (*Prasad et al., 2019*), and, in addition, AS69 or 'always early RNAi' (see Materials and methods section) as a control. Aggregates of α-synuclein were quantified by a filter trap assay in which urea-treated lysates of fly heads were passed through a membrane and the quantity of α-synuclein aggregates retained in the membrane was detected by antibodies raised against α-synuclein (illustrated in *Figure 3c*). We found that the quantity of aggregates retained in the filter was significantly smaller in lysates from flies co-expressing AS69 and A53T-α-synuclein than in lysates from flies only expressing VN- and VC-tagged A53T-α-synuclein (*Figure 3d and e*). These findings confirm that AS69 reduces high molecular weight aggregates of α-synuclein in neuronal cells of *Drosophila melanogaster*.

## AS69 stoichiometrically inhibits the elongation of α-synuclein fibrils

We next set out to elucidate the origin of the remarkable ability of AS69 to inhibit α-synuclein aggregate formation in cells and *in vivo* (*Figure 2*, *Figure 3*), and amyloid fibril formation *in vitro* (*Mirecka et al., 2014*). To this end, we performed a detailed mechanistic analysis, where we examined the effect of AS69 on the growth (*Buell et al., 2014a*), autocatalytic amplification (*Buell et al., 2014a*; *Flagmeier et al., 2016*) and lipid-induced formation (*Galvagnion et al., 2015*) of α-synuclein amyloid fibrils. We first carried out experiments in the presence of micromolar concentrations (in

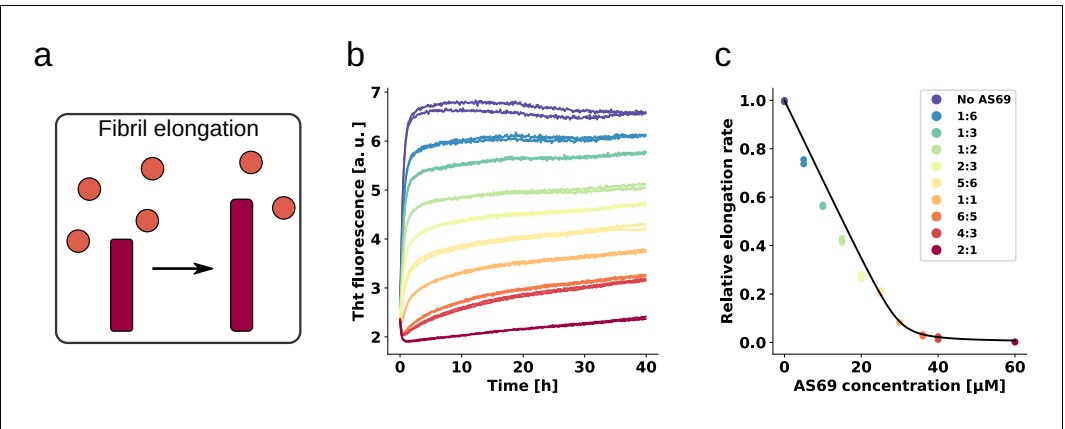

**Figure 4.** AS69 inhibits α-synuclein fibril elongation. (**a**) Schematic representations of fibril elongation. (**b**) Change in ThT fluorescence when a 30 µM solution of monomeric α-synuclein was incubated in the presence of 5 µM pre-formed fibrils under quiescent conditions with increasing concentrations of AS69. (**c**) Relative rates of fibril elongation with increasing concentrations of AS69. The solid line corresponds to a prediction based on the affinity of AS69 for monomeric α-synuclein (240 nM, Figure 1b [*Mirecka et al., 2014*], see Appendix 1 for details). The online version of this article includes the following figure supplement(s) for figure 4:

**Figure supplement 1.** Characterisation of α-synuclein fibrils formed in the presence and absence of AS69 by AFM.

**Figure supplement 2.** Binding specificity determines the inhibitory activity.

monomer equivalents) of pre-formed seed fibrils of α-synuclein at neutral pH under quiescent conditions (*Figure 4a,b*). We have shown previously that under these conditions only fibril elongation through addition of monomeric α-synuclein to fibril ends occurs at detectable rates (*Buell et al., 2014a*), and that the rate of *de novo* formation of fibrils is negligible. We therefore examined the effects of AS69 on fibril elongation and analyzed these data by fitting linear functions to the early stages of the aggregation time courses (see Appendix 1 for details of the analysis). The results indicate that fibril elongation is indeed inhibited by AS69 in a stoichiometric concentration-dependent manner (*Figure 4c*). In this experiment, both the seed fibrils and the AS69 compete for the monomeric α-synuclein and the relative affinities determine the kinetics and thermodynamics of the system.

To obtain an estimate of the affinity of monomeric α-synuclein for the ends of fibrils, we performed elongation experiments at low monomer concentrations in the absence of AS69. We found evidence that the fibrils are able to elongate in the presence of 0.5 µM monomeric α-synuclein (see Appendix 1), providing an upper bound of the critical concentration (which is formally equivalent to a dissociation constant, see Appendix 1). Despite the similar affinity of monomeric α-synuclein for both fibril ends and AS69, the timescales of the two types of interactions are very different; monomeric α-synuclein was found to interact on a timescale of seconds with AS69, as seen by isothermal titration calorimetry (ITC) experiments (*Mirecka et al., 2014* and *Figure 1b and c*), but to incorporate on a timescale of minutes to hours into free fibril ends (see *Figure 4b* and *Buell et al., 2014a*; *Wördehoff et al., 2015*). The slow kinetics of the latter process is partly because the number of fibril ends is much smaller than the number of monomers (*Buell et al., 2014a*), such that each fibril sequentially recruits many α-synuclein molecules. Therefore, the equilibrium between AS69 and α-synuclein should be rapidly established and perturbed only very slowly by the presence of the fibrils.

## Inhibition of fibril elongation is caused by monomer sequestration

The initial fibril elongation rate as a function of AS69 concentration was found to follow closely the predicted concentration of unbound α-synuclein across the entire range of concentrations of AS69 used in this study, as shown in *Figure 4c*, where the solid line corresponds to the predicted elongation rate, assuming fibrils can only be elongated by unbound α-synuclein. The inhibition of fibril elongation can therefore be explained quantitatively by the sequestration of monomeric α-synuclein by AS69 and the assumption that the AS69:α-synuclein complex cannot be incorporated into the

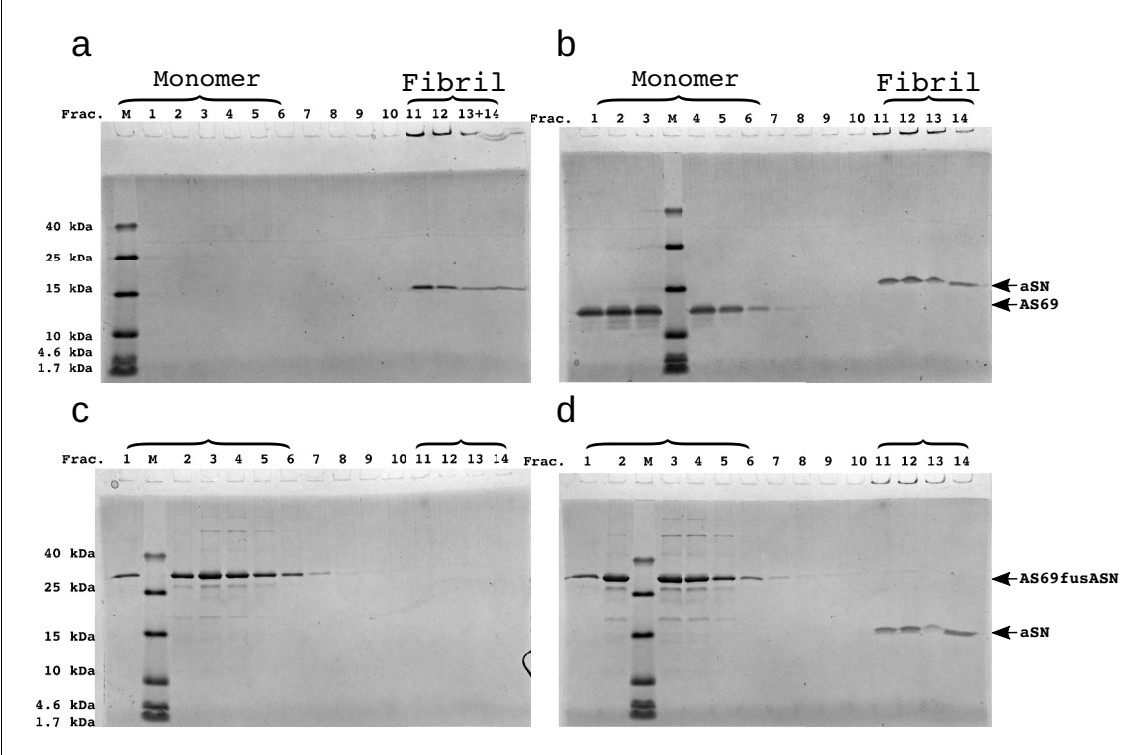

**Figure 5.** SDS-PAGE of density gradient centrifugation (DGC) experiments to probe the binding of AS69 to α-synuclein fibrils at pH 7.4 after elongation experiments. (a) 25 µM seeds, (b) 25 µM AS69 and 25 µM seeds, (c) 16.7 µM AS69fusASN, (d) 25 µM AS69fusASN and 25 µM seeds.

growing fibril. This conclusion is supported by the finding that the fibrils formed in the presence of increasing concentrations of AS69 are morphologically indistinguishable from the fibrils formed in the absence of AS69 (as judged from AFM images, see *Figure 4—figure supplement 1*). Our kinetic analysis of fibril elongation in the presence of AS69 does not, however, suggest a preferential interaction with fibril ends, as such an interaction can be expected to lead to a sub-stoichiometric inhibition of fibril elongation, which is not observed in our experiments. Indeed, the finding that the effect on elongation can be quantitatively described by considering only the interaction of AS69 with monomeric α-synuclein (Appendix 1) suggests a weak, if any, interaction of AS69 with fibrils. Furthermore, density gradient centrifugation (DGC) of samples containing only seeds and AS69 (*Figure 5a and b*) did not show AS69 to co-migrate with large species to any significant extent under conditions that favour elongation. In agreement with inhibition of fibril elongation by monomer sequestration, ZAβ₃W, a binding protein for amyloid-β peptide (*Grüning et al., 2013*), which is a significantly weaker α-synuclein binder than AS69, correspondingly showed a considerably weaker inhibitory effect on α-synuclein fibril elongation (*Figure 4—figure supplement 2*).

## AS69 sub-stoichiometrically inhibits amplification of α-synuclein fibrils

These findings clearly demonstrate that AS69 inhibits fibril elongation in a stoichiometric manner through monomer sequestration. Consequently, inhibition of fibril elongation cannot explain the previously observed sub-stoichiometric inhibition of α-synuclein fibril formation by AS69 (*Mirecka et al., 2014*). We therefore performed seeded experiments under mildly acidic solution conditions in the presence of very low concentrations of pre-formed fibrils (nM monomer equivalents) under quiescent conditions (*Figure 6a,b*) (*Buell et al., 2014a*; *Gaspar et al., 2017*). Under those solution conditions, seeded aggregation has been shown to consist of two processes in addition to fibril elongation, namely secondary nucleation, which increases the number of growth competent fibril ends, and higher order assembly ('flocculation', *Figure 6—figure supplement 1b,c*), which decreases the overall aggregation rate by reducing the number of accessible fibrils through their burial within higher order aggregates (*Buell et al., 2014a*). The de novo formation of amyloid fibrils through primary nucleation is

suppressed if the solution is not agitated and if non-binding surfaces are used (*Figure 6—figure supplement 1a*). We find that under these solution conditions, where only growth and secondary nucleation contribute to the increase in fibril mass and number, respectively, the seeded aggregation is inhibited in a strongly sub-stoichiometric manner (*Figure 6b,c*). We analysed these data to determine the maximum rate of aggregation (see Appendix 2 for details) using the framework from *Cohen et al. (2011)* (*Figure 6c*). Based on recent results on the concentration-dependence of autocatalytic secondary nucleation of $\alpha$-synuclein amyloid fibrils (*Gaspar et al., 2017*), we have calculated the predicted inhibitory effect from monomer sequestration by AS69 in *Figure 6c* (see *Figure 6—figure supplement 2* and Appendix 2 for details). We find that, unlike the case of fibril elongation, monomer sequestration cannot explain the extent of inhibition, even by assuming a very high reaction order of 5 (i.e. a dependence of the rate of secondary nucleation on the 5th power of the free monomer concentration; $\frac{dP(t)}{dt} \propto m(t)^5$) which is not compatible with recent results, showing that secondary nucleation of $\alpha$-synuclein amylid fibrils depends only weakly on the concentration of free monomer (*Gaspar et al., 2017*). However, even in this unlikely scenario, the very strong inhibitory effect of low AS69 concentrations cannot be explained by monomer depletion.

## Sub-stoichiometric inhibition of fibril amplification is not caused by interaction with the fibril surface

We have previously been able to rationalise inhibition of the secondary nucleation of $\alpha$-synuclein by the homologous protein $\beta$-synuclein through competition for binding sites on the surface of the fibrils (*Brown et al., 2016*). Here we find that AS69 is a significantly more efficient inhibitor of the autocatalytic amplification of $\alpha$-synuclein amyloid fibrils than $\beta$-synuclein (a similar degree of inhibition is achieved with a 10-fold lower concentration ratio). This result is particularly interesting in the light of the fact that AS69 binds efficiently to monomeric $\alpha$-synuclein under both neutral and mildly acidic solution conditions (*Figure 1b,c*), whereas we found no evidence for a relevant direct interaction between the monomeric forms of $\alpha$- and $\beta$-synuclein, given the complete absence of any inhibitory effect of $\beta$-synuclein on the elongation of $\alpha$-synuclein fibrils (*Brown et al., 2016*). Therefore, despite the vast majority of the AS69 being bound within a complex with monomeric $\alpha$-synuclein, AS69 is an efficient sub-stoichiometric inhibitor of the secondary nucleation of $\alpha$-synuclein. This

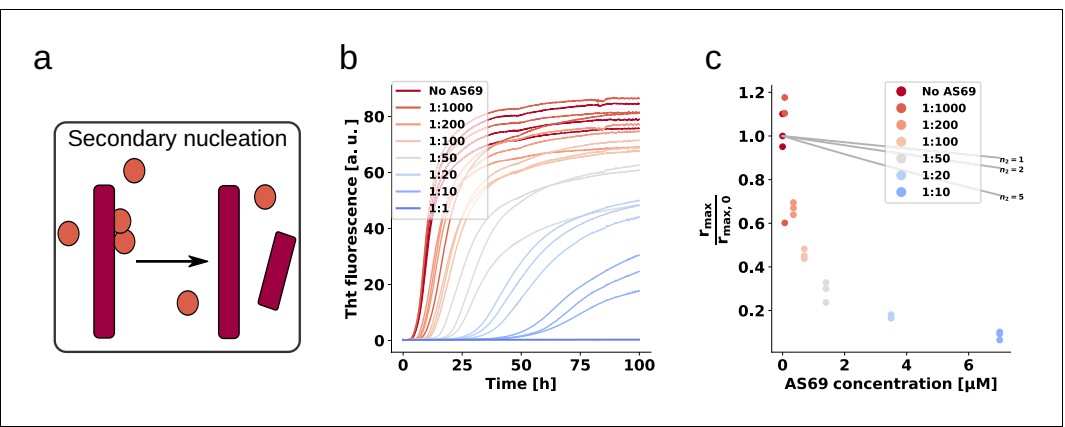

**Figure 6.** AS69 inhibits $\alpha$-synuclein fibril amplification. (a) Schematic representation of fibril amplification through secondary nucleation *Buell et al. (2014a)*. (b) Change in ThT fluorescence intensity when a 70 $\mu$M solution of monomeric $\alpha$-synuclein was incubated with increasing concentrations of AS69 in acetate buffer (pH 5.0) under quiescent conditions and weak seeding. (c) Relative rate of fibril amplification as a function of the concentration of AS69. The solid lines correspond to simulations based on the assumption that AS69 acts only through monomer sequestration, for different values of the monomer dependence (reaction order) of secondary nucleation (see Appendix 2 for details).

The online version of this article includes the following figure supplement(s) for figure 6:

**Figure supplement 1.** Seeds are required for aggregation under quiescent conditions.
**Figure supplement 2.** Weakly seeded aggregation experiments at pH 5.0.
**Figure supplement 3.** AS69 interacts with two distinct $\alpha$-synuclein species.

finding suggests that in addition to inhibiting through competition for nucleation sites on the fibril surface, AS69 or its complex with $\alpha$-synuclein could interact directly with intermediates of the secondary nucleation process. To investigate whether AS69 binds to the fibril surface under these secondary nucleation-inducing solution conditions, we performed additional DGC experiments. Comigration in the density gradient of AS69 with fibrils, which would imply direct interactions between these species, was undetectable (*Figure 7a–c*). If AS69 was able to inhibit secondary nucleation through binding to the fibril surface in the presence of a large excess of monomer, its affinity to fibril surfaces would need to be much higher than to monomeric $\alpha$-synuclein. This implies that under the conditions of the DGC experiments which were performed in the absence of monomeric $\alpha$-synuclein, all binding sites on the fibrils should be occupied. Therefore, the absence of detectable binding implies either a weak affinity for fibrils or a very low stoichiometry, that is a very low density of binding sites for AS69 on the fibril surface.

## AS69 binds to stable $\alpha$-synuclein oligomers with comparable affinity as to monomers

We next tested whether binding of AS69 to oligomeric states of $\alpha$-synuclein could explain the efficient inhibition of secondary nucleation. The heterogeneous and often transient nature of oligomeric intermediates on the pathway to formation of amyloid fibrils makes any interaction between such species and AS69 difficult to probe. However, monomeric $\alpha$-synuclein can be converted into kinetically stable oligomers that can be studied in isolation, because they do not readily convert into amyloid fibrils (*Lorenzen et al., 2014*). Despite it not being likely that these species are fibril precursors, they are intermediate in size and structure between monomeric and fibrillar $\alpha$-synuclein and hence can serve as a model for AS69 binding to $\alpha$-synuclein oligomers. Using microscale thermophoresis (MST, *Wolff et al., 2016*) at neutral pH, we were able to confirm the binding of AS69 to both monomeric (*Figure 6—figure supplement 3a*) and oligomeric $\alpha$-synuclein (*Figure 6—figure supplement 3b*) and provide estimates of the respective binding affinities (ca. 300 nM for monomeric and ca. 30 nM for oligomeric $\alpha$-synuclein). The former value is in good agreement with results from ITC experiments under the same solution conditions (*Figure 1b* and *Mirecka et al., 2014*), whereas the affinity of AS69 to oligomeric $\alpha$-synuclein has not previously been determined. The finding that AS69 is able to inhibit secondary nucleation in a highly sub-stoichiometric manner in the presence of a large excess of free monomer, to which it binds with high affinity, necessitates that the interactions of AS69 with aggregation intermediates must be of significantly higher affinity, if they are to explain the inhibition. Otherwise the monomer would out-compete the aggregation intermediate for AS69 binding, because of the much lower concentration of the latter. An estimate (see Appendix 2 for details) suggests that the affinity of AS69 for aggregation intermediates would need to be several orders of magnitude higher than to $\alpha$-synuclein monomer to explain an inhibitory effect of the observed magnitude. This required affinity is indeed much higher than the affinity we have determined here for an oligomeric state of $\alpha$-synuclein.

## A covalent complex of AS69 and $\alpha$-synuclein efficiently inhibits secondary nucleation

The analysis described in the previous section suggests, therefore, that the $\alpha$-synuclein:AS69 complex itself could be the inhibitory species. The population of this complex is sufficiently high, even at low ratios of AS69:$\alpha$-synuclein, to interact with a considerable fraction of aggregation intermediates. It is possible, therefore, that while the AS69:$\alpha$-synuclein complex is unable to incorporate into a fibril end (see section above on the stoichiometric inhibition of fibril elongation), it can interact with oligomeric fibril precursors and block their conversion into fibrils. We tested this hypothesis by producing a molecular construct whereby $\alpha$-synuclein and AS69 are linked together with a flexible glycine tether that allows formation of an intramolecular complex (AS69fusASN). The formation of the intramolecular complex was verified by performing CD spectroscopy at 222 nm over the temperature range from 10 to 90°C and fitting the data to a two-state model (*Pace et al., 1998*) (see *Figure 8—figure supplement 1*). Both at neutral and mildly acidic pH, the fusion construct AS69fusASN has a higher thermal stability than the free AS69 and, indeed, as the stoichiometric mixture of AS69 and $\alpha$-synuclein (*Table 1*). The difference in melting temperatures between the covalent and non-covalent complex can be explained by the differences in the entropy of binding, which is more

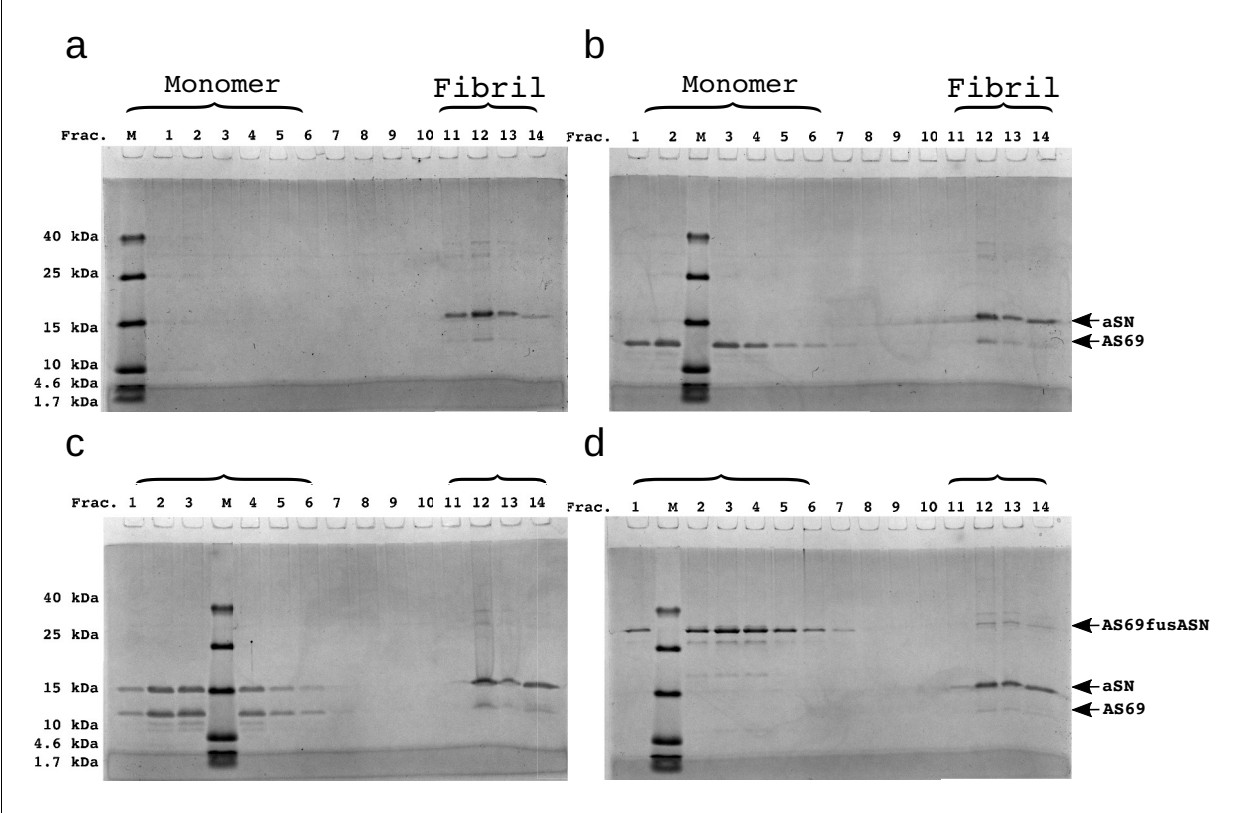

**Figure 7.** SDS-PAGE of density gradient centrifugation experiments to probe for binding of AS69 to fibril surfaces at pH 5.0. (a) 12.5 µM seeds, (b) 12.5 µM AS69 and 12.5 µM seeds, (c) 12.5 µM AS69, 12.5 µM seeds and 12.5 µM monomer, and (d) 12.5 µM AS69fusASN and 12.5 µM seeds.

unfavourable in the case of the non-covalent complex, given the loss of three degrees of freedom of translational motion upon binding.

We performed weakly seeded aggregation experiments under conditions where secondary nucleation leads to the amplification of the added seed fibrils (see above) at different concentrations of AS69 (*Figure 8a,c*), as well as AS69-α-syn complex (*Figure 8b,d*) We found that the pre-formed complex is a similarly efficient inhibitor as the free AS69 under secondary nucleation conditions (*Figure 8e*). These results provide strong support for our hypothesis that the AS69-α-synuclein complex, covalent or non-covalent, is the species that is responsible for the sub-stoichiometric inhibition of secondary nucleation. Therefore, we propose a model whereby rather than requiring the binding of free AS69 to an aggregation intermediate, the AS69:α-synuclein complex is able to incorporate into a fibril precursor and efficiently prevent it from undergoing the structural rearrangement required to transform into a growth-competent amyloid fibril.

## AS69 inhibits lipid-induced aggregation of α-synuclein

Having established and rationalised the high efficiency of AS69 to inhibit autocatalytic amplification of α-synuclein amyloid fibrils through secondary nucleation, we next investigated whether the *de novo* formation of α-synuclein amyloid fibrils is also efficiently inhibited. As experimental setup, we chose a recently developed paradigm of lipid-induced aggregation (*Galvagnion et al., 2015*), which allows analysis of the resulting kinetic data in a more quantitative manner compared to the widely employed conditions of strong mechanical agitation and high affinity multiwell plate surfaces. In the latter conditions, the dominant role of the air-water interface (*Campioni et al., 2014*) as well as of fragmentation have rendered challenging quantitative analysis of the resulting data. In the lipid-induced aggregation, under quiescent conditions and in non-binding plates, the nucleation on the lipid vesicles is the dominant source of new α-synuclein amyloid fibrils. We therefore probed the

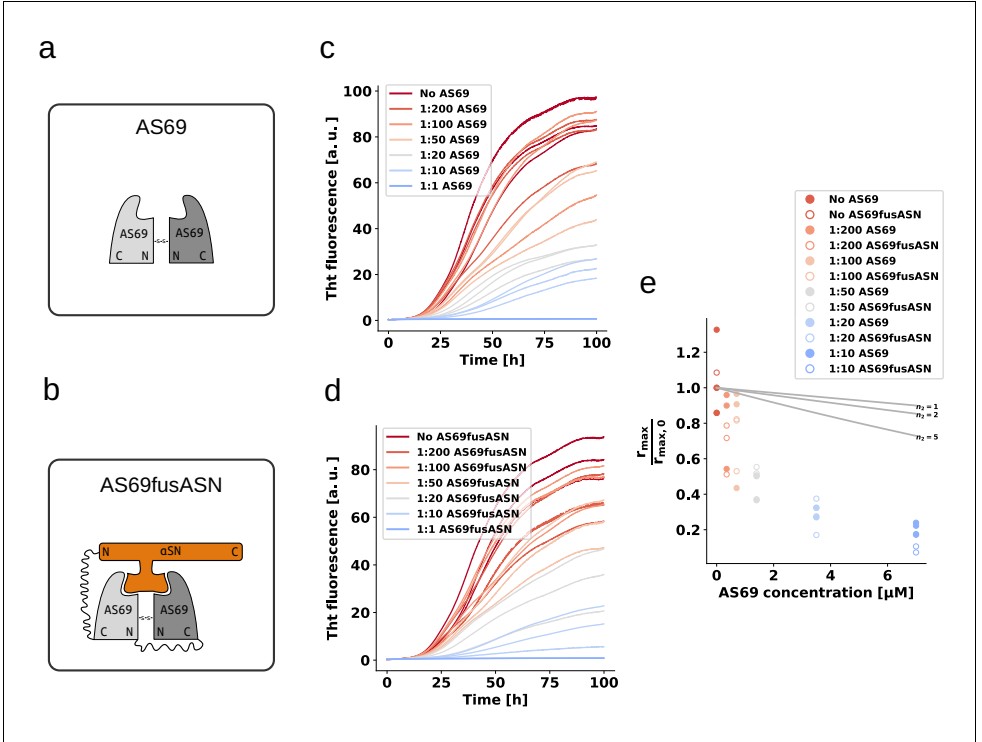

**Figure 8.** AS69 and AS69fusASN inhibit α-synuclein fibril amplification to similar extent. (**a**) and (**b**) Schematic representations of AS69 and AS69fusASN, respectively. (**c**), (**d**) Change in ThT fluorescence when a 70 µM solution of monomeric α-synuclein was incubated with increasing concentrations of AS69 or AS69fusASN, respectively, in sodium acetate buffer (pH 5.0) under quiescent conditions. (**e**) Relative maximum rate of aggregation as a function of the concentration of AS69 (closed circles) and AS69fusASN (open circles). The solid lines correspond to simulations based on the assumption that AS69 acts only through monomer sequestration, for different values of the monomer dependence (reaction order) of secondary nucleation (see Appendix 2 for details).

The online version of this article includes the following figure supplement(s) for figure 8:

**Figure supplement 1.** Determination of thermal stabilities of AS69 and its non-covalent and covalent complex with α-synuclein.

**Figure supplement 2.** Weakly seeded aggregation experiments at mildly acidic pH 5.

inhibitory effect of AS69 on lipid vesicle (DMPS-SUV)-induced aggregation of α-synuclein (**Figure 9a, b**). We then analysed the early times of the kinetic traces using a single-step nucleation model (**Figure 9c**) that includes only primary nucleation and fibril elongation (see Appendix 3). The results reveal that AS69 inhibits lipid-induced aggregation at sub-stoichiometric concentrations to α-synuclein in a concentration-dependent manner (**Figure 9c**). To characterise the system α-synuclein-AS69-DMPS-SUV in more detail, we performed titration experiments where we varied the concentration of SUVs at constant α-synuclein:AS69 ratios of 10:1 and 1:1. We monitored the formation of α-helical structure, induced by binding of α-synuclein to the DMPS-SUV by circular dichroism (CD) spectroscopy (**Figure 9—figure supplement 1a–c**). We find that the system is well-described as a competition between the AS69 and the lipid vesicles for the monomeric α-synuclein (**Figure 9—figure supplement 1d** and see Materials and methods section for details of the mathematical analysis). We simulated the effects that AS69 has on the aggregation process of α-synuclein in the presence of lipids, assuming that sequestration of free monomer is the only mechanism through which AS69 inhibits the aggregation reaction (**Figure 9c**). The results show that the lipid-induced aggregation of α-synuclein is inhibited by AS69 significantly more strongly than predicted by monomer sequestration alone. However, before being able to conclude that AS69 inhibits the lipid-induced aggregation of α-synuclein through a mechanism similar to that defined above for secondary nucleation, it needs

**Table 1.** Melting temperatures, $T_m$, obtained from fitting of CD melting curves in **Figure 8—figure supplement 1**.
*Data from **Gauhar et al. (2014)** was refitted to obtain the numerical values listed in the table.

| Construct | $T_M$ [°C] at pH 7.4 | $T_M$ [°C] at pH 5 |
|---|---|---|
| AS69 | 37.5(± 1.6)* | *36.5(± 1.8) |
| AS69 + α-synuclein | 51.0(± 0.6)* | 55.8(± 0.2) |
| AS69fusASN | 66.5(± 0.3) | 66.1 (± 0.2) |

to be established whether or not AS69 can directly interact with the lipid vesicles and exert an inhibitory effect through this interaction. We have previously reported that this type of inhibition is displayed by β-synuclein, a homologous protein which directly competes with α-synuclein for binding sites on the lipid vesicles (**Brown et al., 2016**). To test for a direct interaction between AS69 and the DMPS-SUV, we performed both isothermal titration and differential scanning calorimetry (ITC and DSC, **Figure 9—figure supplement 2**). We find that the melting temperature of DMPS vesicles is decreased in the presence of AS69 (**Figure 9—figure supplement 2a,b**) and, furthermore, titration of AS69 into DMPS-SUV reveals a complex signature of heat release and consumption (**Figure 9—figure supplement 2c,d**). While a detailed analysis of this interaction behaviour is beyond the scope of the present study, taken together these calorimetric experiments suggest indeed a direct interaction between AS69 and DMPS-SUV. Therefore, despite AS69 appearing to be a more potent inhibitor of lipid-induced aggregation than β-synuclein, with similar inhibitory effects for very different ratios of inhibitor to α-synuclein of 5:1 (β-synuclein) and 1:10 (AS69), it cannot be excluded that the same mechanism of inhibition contributes significantly to the overall inhibitory effect in lipid-induced aggregation.

## Discussion

The β-wrapin AS69 is a small engineered monomer binding protein that upon coupled folding-binding induces a local β-hairpin conformation in the region comprising amino acid residues 37–54 of otherwise intrinsically disordered monomeric α-synuclein (**Figure 1**). AS69 shows strongly sub-stoichiometric inhibition of α-synuclein aggregation in vitro, which is remarkable for a monomer binding-protein (**Mirecka et al., 2014**). Here, we show that potent aggregation inhibition of AS69 can be recapitulated in cell culture as well as an animal model. In cell culture, AS69 interfered with the interaction between tagged α-synuclein molecules as judged by a fluorescence complementation assay and reduced the formation of visible aggregate particles of GFP-tagged α-synuclein (**Figure 2**). In fruit flies, co-expression of AS69 led to reduced abundance of large molecular weight aggregates of tagged α-synuclein and rescue of the motor phenotype resulting from neuronal expression of untagged A53T-α-synuclein (**Figure 3**). While the nature of the α-synuclein aggregates formed inside the cells and fly neurons remains elusive, these results show that AS69 is able to interact with different constructs and forms of α-synuclein in vivo, and hence its inhibition of α-synuclein amyloid fibril formation observed in vitro (**Mirecka et al., 2014**) warrants further in-depth analysis. Our detailed biophysical in vitro aggregation experiments under well-defined conditions enabled us to reveal several distinct modes of inhibition of α-synuclein amyloid fibril formation by AS69, as summarised in **Figure 10**. First, as expected for a monomer-binding species, AS69 inhibits fibril growth in a strictly stoichiometric manner, suggesting that the non-covalent AS69-α-synuclein complex is unable to add onto a fibril end and elongate the fibril. This is consistent with our results from DGC regarding the lack of a detectable interaction between AS69 and fibrils. Second, AS69 is found to be a very efficient inhibitor of secondary nucleation at highly sub-stoichiometric ratios. The overall result of our experimental and theoretical analysis is that this inhibitory effect is unlikely to stem from a direct interaction between the AS69 and either fibril surfaces or secondary nucleation intermediates. Such an interaction would need to be of an unrealistically higher affinity than the interaction between AS69 and α-synuclein monomer. A possible solution to this conundrum is presented by the hypothesis that the AS69-α-synuclein complex is the inhibitory species. This hypothesis gains strong support

from our finding that a covalently linked complex is equally as efficient an inhibitor of secondary nucleation as the free AS69 molecule. It is important to note here that this proposed mode of action is very distinct from other types of inhibitory behavior reported previously. For example in the case of nanobodies raised against monomeric α-synuclein, at least stoichiometric amounts of the nanobodies are needed to interfere significantly with unseeded aggregation (*Iljina et al., 2017*). In the case of molecular chaperones, on the other hand, sub-stoichiometric inhibitory behaviour has been reported previously (*Waudby et al., 2010*; *Månsson et al., 2014*), but it is usually found that these molecules do not interact significantly with the monomer, but rather bind specifically to aggregated states of the protein. Therefore, the AS69 affibody represents a new paradigm in the inhibition of amyloid fibril formation: strongly sub-stoichiometric inhibition by a tight monomer-binding species. In this scenario, it is not the inhibitor itself that plays the role of a molecular chaperone, that is interacting with an on-pathway species and interfering with its further evolution, but rather the monomer-inhibitor complex acts as a chaperone. This mode of action represents a range of significant advantages over the other previously described modes of action (i.e. monomer sequestration and direct interaction with aggregation intermediates). First, it is rather straightforward to develop further molecules that bind to the monomeric forms of proteins, given that the latter are well-defined, reproducible and easy to handle. This simplicity is in contrast to the difficulty presented by targeting on-pathway aggregation intermediates which are difficult to isolate for the development of inhibitors. Second, binders of oligomeric aggregation intermediates can be expected to be less specific compared to binders of a well-defined monomeric state, as suggested by the existence of antibodies that interact with protofibrillar species independently of the protein from which they have formed (*Kayed et al., 2003*). This lack of specificity can potentially lead to cross-reactivity and side effects. And third, the mode of inhibition presented here avoids the need for stoichiometric amounts of inhibitors that are usually required in the case of monomer-sequestering species, resulting in a more efficient inhibition. Interestingly, we find that AS69 is a similarly potent inhibitor in a lipid-induced aggregation paradigm, whereby heterogeneous primary, rather than secondary, nucleation is the dominant source of new aggregates. However, we found the inhibitory effect in this case possibly

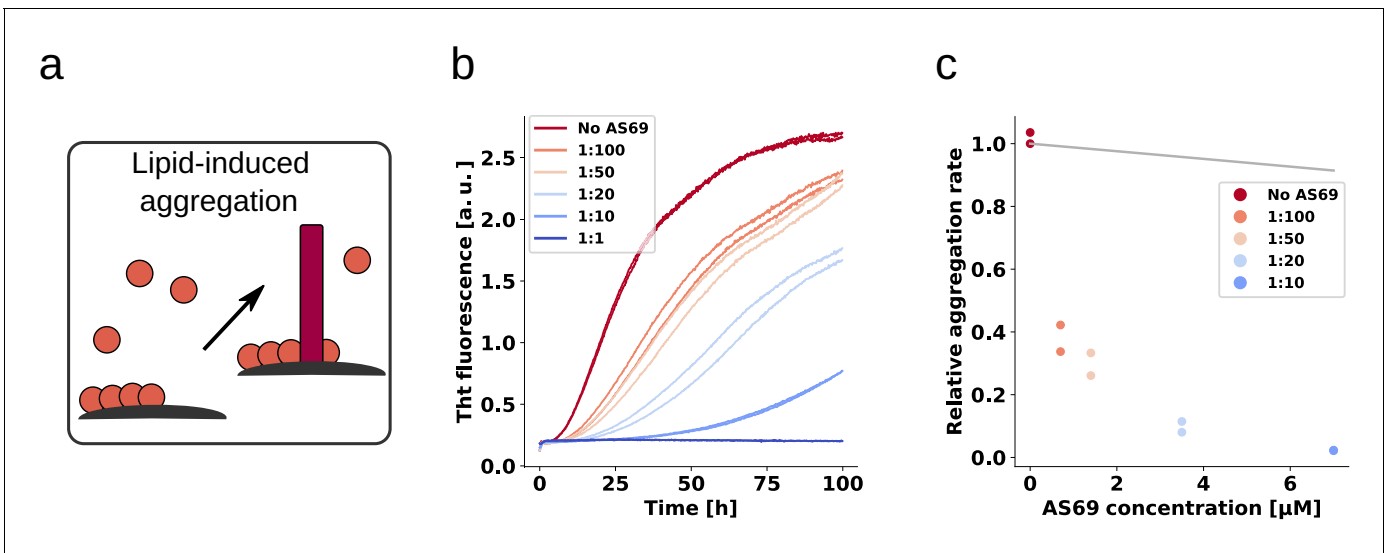

**Figure 9.** AS69 inhibits lipid-induced aggregation of α-synuclein. (a) Schematic representation of lipid-induced aggregation (*Galvagnion et al., 2015*). (b) Change in ThT fluorescence intensity when a 70 μM solution of monomeric α-synuclein was incubated with 100 μM DMPS-SUVs and increasing concentrations of AS69 in 20 mM phosphate buffer (pH 6.5) under quiescent conditions. (c) Relative rate of lipid-induced formation of α-synuclein amyloid fibrils as a function of the concentration of AS69. The solid line corresponds to a simulation based on the assumption that AS69 acts only through monomer sequestration (see Appendix 3 for details).

The online version of this article includes the following figure supplement(s) for figure 9:

**Figure supplement 1.** Influence of AS69 on the lipid-binding of α-synuclein monitored using circular dichroism.
**Figure supplement 2.** Calorimetric experiments designed to elucidate the molecular mechanism of inhibition of lipid-induced aggregation of α-synuclein by AS69.

also stemmed from a direct interaction between AS69 and the lipid vesicles. It is therefore not straightforward to decide whether the dominant mechanism of inhibition by AS69 in heterogeneous primary and secondary nucleation is closely related.

An inhibitor functioning according to this dual mode, that is being active both as a free molecule and as a complex with monomeric $\alpha$-synuclein, is expected to efficiently reduce $\alpha$-synuclein aggregation in vivo. This is in agreement with the cell culture and fly data we present in this manuscript. Further steps will be to test the effects of AS69 in cell-based fibril seeding assays, in mammalian dopaminergic neurons, and in PD models where synuclein aggregates are formed from endogenous $\alpha$-synuclein.

In conclusion, high affinity monomer binders displaying strong sub-stoichiometric inhibition of fibril formation represent attractive agents to interfere with pathological protein aggregation, as a result of their multiple inhibitory action.

# Materials and methods

## Reagents

Thioflavin T UltraPure Grade (ThT > 95%) was purchased from Eurogentec Ltd (Belgium). Sodium phosphate monobasic (NaH$_2$PO$_4$, BioPerformance Certified >99.0%), sodium phosphate dibasic (Na$_2$HPO$_4$, ReagentPlus, >99.0%) and sodium azide (NaN$_3$, ReagentPlus, >99.5%) were purchased from Sigma Aldrich, UK. 1,2-Dimyristoyl-sn-glycero-3-phospho-L-serine, sodium salt (DMPS) was purchased from Avanti Polar Lipids, Inc, USA.

## Protein preparation

$\alpha$-synuclein was expressed and purified as described previously (*Hoyer et al., 2002*; *Buell et al., 2014a*). To determine the concentrations in solution, we used the absorbance value of the protein measured at 275 nm and an extinction coefficient of 5600 M$^{-1}$cm$^{-1}$. The protein solutions were divided into aliquots, flash-frozen in liquid N$_2$ and stored at $-80$°C, until used. A pET302/NT-His plasmid carrying AS69 with a N-terminal hexahistag (on each monomer) was expressed and purified as previously described (*Mirecka et al., 2014*) in *E. coli* JM109(DE3) with small modifications. Briefly, 20 µl cell culture from a glycerol stock was used to inoculate 50 ml 2YT (PanReac AppliChem) with 100 µg / ml ampicillin overnight culture, from which 5 ml was added per 500 ml 2YT medium with 100 µg / ml ampicillin. Expression was induced when OD600 reached 0.6, using IPTG to a final concentration of 1 mM, after which the cells were grown for an additional 4 h; the temperature of growth and expression was 37°C and shaking was 110 RPM. Cells were harvested by centrifugation at 5000 g for 20 min at 4°C, after which the cell pellets were resuspended in 50 mM Tris:Cl pH 8, 500 mM NaCl, 20 mM imidazole, and one protease inhibitor cocktail tablet (Roche) before being placed at $-20$°C. Cells were thawed and lysed using a probe sonicator (Bandelin, Sonopuls UW 3200, Berlin, Germany) with a MS72 sonotrode, with pulses of 3 s with pauses of 5 s in between for a

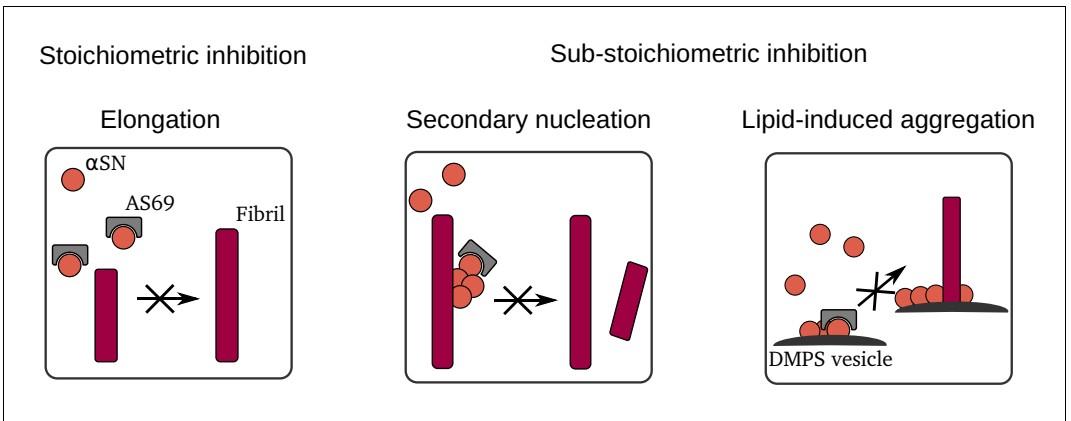

**Figure 10.** Summary of mechanisms by which AS69 inhibits amyloid fibril formation of $\alpha$-synuclein in vitro.

total of 5 min using 35% maximum power. Cell debris was removed by centrifugation at 13500 g for 20 min, before the supernatant was loaded on a 5 ml Histrap FF (GE Healtcare). A 50 mM imidazole-containing buffer (as opposed to 20 mM; see above) was loaded to remove unspecifically bound material before elution was performed using 250 mM imidazole. The eluate was placed on ice over-night before it was concentrated to a volume < 2.5 ml and then loaded onto a Hiload 16/600 Super-dex 75 pg column, that had been equilibrated in 20 mM NaPi, pH 7.4, 50 mM NaCl, for collection of the dimer peak. Protein concentration was measured at 275 nm with an extinction coefficient of 2800 $M^{-1}$ $cm^{-1}$, protein solutions were aliquoted, flash-frozen in in liquid $N_2$ and stored at $-80°C$. AS69fusASN with a C-terminal hexahistag was expressed from a pET302/CT-His plasmid and puri-fied identically to AS69 with the only exception that an anion exchange chromatography step was included (identical to the one used for $\alpha$-synuclein). Protein concentration was measured at 275 nm with an extinction coefficient of 8400 $M^{-1}$ $cm^{-1}$, protein solutions were aliquoted, flash-frozen in in liquid $N_2$ and stored at $-80°C$.

## Seed fibril formation

Seed fibrils were produced under different solution conditions, depending on which type of experi-ments they were needed for (see section on ThT experiments below).

### Elongation assays

Seed fibrils were produced as described previously (*Buell et al., 2014a*). 500 µl samples of $\alpha$-synu-clein at concentrations from 500 to 800 µM were incubated in 20 mM phosphate buffer (pH 6.5) for 48–72 h at ca. 40°C and stirred at 1500 rpm with a Teflon bar on an RCT Basic Heat Plate (IKA, Stau-fen, Germany). Fibrils were diluted to a monomer equivalent concentration of 200 µM, divided into aliquots, flash-frozen in liquid $N_2$ and stored at $-80°C$. For experiments at pH 6.5 and 5 µM fibril concentrations, the 200 µM fibril stock was sonicated between 30 s and 1 min using a probe sonica-tor (Bandelin, Sonopuls HD 2070, Berlin, Germany), using 10% maximum power and a 50% cycle.

### Secondary nucleation assays

Seed fibrils were produced in 10 mM acetate buffer at pH 5.0. A 1.2 ml sample of $\alpha$-synuclein at a concentration of 25 µM was prepared and aliquoted into 12 wells of a 96-well Half Area Black Flat Bottom Polystyrene NBS Microplate (Corning), where a single glass bead of 2.85–3.45 mm diameter (Carl Roth) had been added. The plate was incubated at 37°C for 48–72 h at 500 RPM. Sonication was performed using a probe sonicator (Bandelin, Sonopuls UW 3200, Berlin, Germany) with a MS72 sonotrode five times for 1 s using 10% maximum power.

## Lipid vesicle preparation

DMPS lipid powder was dissolved in 20 mM phosphate buffer ($NaH_2PO_4/Na_2HPO_4$), pH 6.5, 0.01% $NaN_3$ and stirred at 45°C for at least 2 h. The solutions were then frozen and thawed five times using dry ice and a water bath at 45°C. Lipid vesicles were prepared by sonication (Bandelin, Sonopuls HD 2070, 3 × 5 min, 50% cycle, 10% maximum power) and centrifuged at 15000 rpm for 30 min at 25°C. The average size of the vesicles was verified by dynamic light scattering (Zetasizer Nano ZSP, Mal-vern Instruments, Malvern, UK) to ensure a distribution centred at a diameter of 20 nm.

## Circular dichroism (CD) measurements and data analysis of $\alpha$-synuclein - lipid interactions in the presence of AS69

Samples were prepared as described before (*Galvagnion et al., 2015*) by incubating 20 µM $\alpha$-synu-clein with 2 or 20 µM AS69 and DMPS concentrations ranging from 0 to 1.2 mM in 20 mM phos-phate buffer, pH 6.5, 0.01% $NaN_3$. Far-UV CD spectra were recorded on a JASCO J-810 instrument (Tokyo, Japan) equipped with a Peltier thermally controlled cuvette holder at 30 °C. Quartz cuvettes with path lengths of 1 mm were used, and the CD signal was measured at 222 nm by averaging 60 individual measurements with a bandwidth of 1 nm, a data pitch of 0.2 nm, a scanning speed of 50 nm/min and a response time of 1 s. The signal of the buffer containing DMPS and different concen-trations of AS69 was subtracted from that of the protein. The data were then analysed as described previously (*Galvagnion et al., 2015*; *Brown et al., 2016*). First the fraction of protein bound to

DMPS for the different [$\alpha$-synuclein], [DMPS] and [AS69] used in our study was determined using the following equation:

$$x_b = \frac{CD_{mes} - CD_{free}}{CD_{bound} - CD_{free}} \qquad (1)$$

where $CD_{free}$ is the signal of $\alpha$-synuclein measured in the absence of both DMPS and AS69, $CD_{bound}$ is the signal of the $\alpha$-synuclein measured in the presence of DMPS only under saturating conditions and $CD_{mes}$ is the signal of the $\alpha$-synuclein measured at a given [DMPS] and [AS69].

The values of $x_b$ obtained from our CD measurements were then compared to those estimated from a competitive binding model where both AS69 and DMPS compete for the binding to $\alpha$-synuclein molecules using the binding constants of the systems AS69:$\alpha$-synuclein and DMPS:$\alpha$-synuclein, determined from previous studies (*Mirecka et al., 2014*; *Galvagnion et al., 2015*). We considered the following two equilibria:

$$\alpha + (DMPS)_L \rightleftharpoons \alpha(DMPS)_L$$

$$\alpha + AS69 \rightleftharpoons \alpha\,AS69$$

that are described by the following equations:

$$K_{D,\alpha-DMPS} = \frac{[DMPS_f][\alpha_f]}{L_\alpha[\alpha_b]} \qquad (2)$$

$$K_{D,\alpha-AS69} = \frac{[\alpha_f][AS69_f]}{[AS69_b]} \qquad (3)$$

with

$$[\alpha] = [\alpha_f] + [\alpha_b] + [AS69_b] \qquad (4)$$

$$[DMPS] = [DMPS_f] + L_\alpha[\alpha_b] \qquad (5)$$

$$[AS69] = [AS69_f] + [AS69_b] \qquad (6)$$

where $K_{D,\alpha-DMPS}$, $K_{D,\alpha-AS69}$ are the binding constants of the system DMPS:$\alpha$-synuclein and AS69:$\alpha$-synuclein, respectively; $L_\alpha$ is the stoichiometry in which DMPS binds to $\alpha$-synuclein, that is the number of DMPS molecules interacting with one molecule of $\alpha$-synuclein; $[\alpha]$,$[\alpha_f]$, $[\alpha_b]$ are the concentrations of total, free and DMPS-bound $\alpha$-synuclein; [AS69], $[AS69_f]$, $[AS69_b]$ are the concentrations of total, free and $\alpha$-synuclein-bound AS69; and [DMPS] and $[DMPS_f]$ are the concentrations of total and free $\alpha$-synuclein. The change in the fraction of protein bound with increasing concentration of DMPS can be described using the standard solution of the cubic equation:

$$K_{D,\alpha-DMPS} = \frac{([DMPS] - L_\alpha[\alpha_b])([\alpha] - [\alpha_b] - [AS69_b])}{[\alpha_b]L_\alpha} \qquad (7)$$

$$[AS69_b] = \frac{[AS69] - [\alpha_b] + [\alpha] + K_{D,\alpha-AS69} - \sqrt{4([\alpha_b][AS69] - [AS69][\alpha]) + ([AS69] - [\alpha_b] + [\alpha] + K_{D,\alpha-AS69})^2}}{2[\alpha]}$$

Its solution is not shown here because of its length. For each data point, the concentrations $[\alpha_b]$, [AS] and [DMPS] are known and the equilibrium constants and stoichiometry for the $\alpha$-synuclein: DMPS and $\alpha$-synuclein:AS69 systems were set to the values determined previously (*Galvagnion et al., 2015*; *Mirecka et al., 2014*).

## DSC and ITC measurements

DSC experiments with lipid vesicles, α-synuclein and AS69 (*Figure 9—figure supplement 2a and b*) were performed as described previously (*Galvagnion et al., 2016*). We used a VP-DSC calorimeter (Malvern Instruments, Malvern, UK) at a scan rate of 1˚C per minute. The lipid concentration was 1 mM and the protein concentrations are indicated in the figure legend.

ITC binding experiments between AS69 and α-synuclein were performed on a Microcal iTC200 calorimeter (GE Healthcare) at 30˚C. The buffer was either 20 mM sodium phosphate, 50 mM NaCl, pH 7.4, or 20 mM sodium acetate, pH 5.0. AS69 was used as titrant in the cell at a concentration of approximately 40 μM, and α-synuclein at approximately 10-fold higher concentration as titrant in the syringe. The heat of post-saturation injections was averaged and subtracted from each injection to correct for heats of dilution and mixing. Data were processed using MicroCal Origin software provided with the calorimeter. Dissociation constants were obtained from a nonlinear least-squares fit to a 1:1 binding model.

ITC binding experiments between SUVs made from DMPS and AS69 ( *Figure 9—figure supplement 2c and d*) were performed using an ITC200 instrument (Malvern Instruments, Malvern, UK). A solution of 0.47 mM AS69 was titrated into 0.5 mM DMPS in 20 mM phosphate buffer pH 6.5 at 30˚C, corresponding to the conditions under which the lipid-induced aggregation of α-synuclein had been studied. An interaction between AS69 and DMPS vesicles can be clearly detected, and the binding behaviour is complex, with an initially exothermic interaction at low protein to lipid ratios, followed by an endothermic interaction at molar ratios higher than 0.05. Because of the complex binding signature, it is not straightforward to fit the data and extract a binding affinity but it can be estimated that the binding affinity is in the sub-micromolar range, comparable to that of α-synuclein to the same lipid vesicles (*Galvagnion et al., 2015*).

## Thioflavin-T (ThT) fluorescence assays of amyloid formation kinetics

The ThT experiments were performed under two distinct sets of solution conditions. Firstly, we used phosphate buffer (PB) at pH 6.5, where we have previously shown that highly quantitative kinetic data of amyloid fibril growth can be obtained, and where under strongly seeded and quiescent conditions, all nucleation processes can be neglected (*Buell et al., 2014a*). Furthermore, we also employed mildly acidic solution conditions (acetate buffer at pH 5.0), where secondary nucleation is strongly enhanced and can be conveniently studied (*Buell et al., 2014a*; *Gaspar et al., 2017*). In most of the ThT experiments, samples of 100 μl were loaded into a 96-well Half Area Black Flat Bottom Polystyrene NBS Microplate (Corning, product number 3881). 150 μl of water was added into the wells directly surrounding the wells containing sample, and the outer most wells were not used for experimental measurements. These measures minimise sample evaporation during prolonged kinetic experiments. The plate was sealed using clear sealing tape (Polyolefin Acrylate, Thermo Scientific) and placed inside a platereader (CLARIOStar or FLUOStar Omega, BMG LABTECH, Germany) that had been equilibrated to 37˚C. Data points were obtained every 120–360 s, depending on the duration of the experiment. In some experiments, the fluorescence was read by averaging 12–20 points, measured in a ring with a diameter of 3 mm (orbital averaging mode). Excitation and emission in the CLARIOStar (monochromator) was 440 nm (15 nm bandwidth) and 485 nm (20 nm bandwidth), respectively. Excitation and emission in the FLUOStar Omega (filter) was 448 nm (10 nm bandwidth) and 482 nm (10 nm bandwidth), respectively. In addition to the proteins of interest and buffer, all samples contained 0.04% (w/v) $NaN_3$ and 40 or 50 μM ThT.

## Preparation of fluorescently labelled oligomers

Fluorescently labelled α-synuclein oligomers were prepared as described previously (*Pinotsi et al., 2014*; *Wolff et al., 2016*). In brief, we produced fluorescently labelled α-synuclein monomer by expressing and purifying the N122C cystein variant of α-synuclein, which was then labelled through an incubation with a 10-fold excess of Alexa 647 maleimide (Thermo Fisher Scientific, Loughborough, UK), followed by removal of the excess dye with a Superdex 200 10/300 Increase gel filtration column (GE Healthcare, Amersham, UK). Wild-type and fluorescently labeled N122C variant α-synuclein were combined at a ratio of 30:1, corresponding approximately to the stoichiometry of the oligomers (*Lorenzen et al., 2014*), at a total concentration of ca. 200 μM, dialysed against distilled water for 24 h and lyophilised. The dry protein was redissolved in PBS at concentrations between

500 and 800 µM and incubated at RT overnight under quiescent conditions. The oligomers were then separated from the monomeric protein and larger aggregates using a Superdex 200 10/300 Increase column that had been equilibrated with 20 mM phosphate buffer pH 7.4 and 50 mM NaCl, collecting fractions of 500 µl. The exact concentrations of the oligomer fractions are difficult to determine, because of the weak absorption signal. However, based on the absorptions at 275 nm and 647 nm, we estimated the oligomer concentration to be 3–6 µM in monomer equivalents, corresponding to an oligomer number concentration of 100–200 nM, which also corresponds roughly to the concentration of Alexa label.

## AFM images

### pH 6.5
Atomic force microscopy images were taken with a Nanowizard II atomic force microscope (JPK, Berlin, Germany) using tapping mode in air. Solutions containing fibrils were diluted to a concentration of 1 µM (in monomer equivalents) in water and 10 µl samples of the diluted solution were deposited on freshly cleaved mica and left to dry for at least 30 min. The samples were carefully washed with ~50 µl of water and then dried again before imaging.

### pH 5
Atomic force microscopy images were taken with a Bruker Mulitmode 8 (Billerica, Massachusetts, USA) using ScanAsyst-Air cantilvers (Camarillo, California, USA) using the ScanAsyst PeakForce tapping in air. 15 µl of a 0.7 µM fibril-containing solution was deposited on freshly cleaved mica and incubated for 10 min before the sample was carefully rinsed by applying and removing 100 µl water three times before the sample was dried under a gentle stream of nitrogen.

## DGC

The DGC experiments were performed as previously described (*Rösener et al., 2018*). We performed DGC experiments both under conditions of neutral pH (pH 7.4), where the reaction is elongation dominated, and under mildly acidic conditions (pH 5.0,) where secondary nucleation strongly contributes to the reaction. We find that under both sets of conditions there is no detectable binding between amyloid fibrils and AS69.

## Thermophoresis experiments

The thermophoresis experiments with fluorescently labeled monomeric and oligomeric α-synuclein were performed as described previously (*Wolff et al., 2016*), using a Monolith instrument (Nanotemper, Munich, Germany) and glass capillaries (Nanotemper, Munich, Germany) with hydrophobic coating (oligomeric α-synuclein) or uncoated (monomeric α-synuclein). A two-fold dilution series of AS69 in 20 mM phosphate buffer pH 7.4 with 50 mM NaCl was prepared and then either 10 µl of 5x diluted oligomers (corresponding to 0.6–1.2 µM) or 1 µM labelled monomer was added to each sample of the dilution series. We performed the binding experiments under these buffer conditions for optimal comparability with previous ITC experiments of AS69 binding to monomeric α-synuclein (*Mirecka et al., 2014*).

MST experiments were performed at 40% laser power and 75% LED power (oligomers) or 60% laser power and 20% LED power (monomers). For calculation of the relative change in fluorescence from thermophoresis, the cursors were set before the temperature jump followed by 5 s after the temperature jump (oligomers) and 45 s after the temperature jump (monomers).

## CD melting curves

CD melting curves were obtained as described in *Gauhar et al. (2014)*, with the sole difference that slightly higher concentrations of protein were used, and the samples were heated to 90°C rather than 80°C. The CD data were fitted directly using a two-state model to obtain the melting temperature, $T_m$, as described in *Pace et al. (1998)*:

$$y = \frac{(y_f + m_f T) + (y_u + m_u T) \cdot \exp\left(\frac{\Delta H_m}{RT} \cdot \frac{T - T_m}{T_m}\right)}{1 + \exp\left(\frac{\Delta H_m}{RT} \cdot \frac{T - T_m}{T_m}\right)} \tag{8}$$

using least-square fitting from the Python packages `scipy.optimize.curve_fit`. $y$ is the CD signal in mdeg, $y_f + m_f T$ and $y_u + m_u T$ describes linear change in CD signal of the folded and unfolded state with respect to temperature, respectively, $T$ is the temperature in Kelvin, $R$ is the ideal constant constant, and $\Delta H_m$ is the change in enthalpy at $T_m$.

## Cell culture and transfections

HEK293 cells (RRID CVCL0063) were obtained from the Department of Biochemistry, RWTH Aachen University, Aachen, Germany, and were cultured and transfected using Metafectene as previously described (*Dinter et al., 2016*). Cell line authentication was performed by Eurofins Forensik, using PCR-single-locus-technology. Cell lines were tested for mycoplasma contamination. HEK293T cells were used because they are the established cell line for our protocol. A53T-$\alpha$-synuclein flexibly tagged with EGFP by the interaction of a PDZ domain with its binding motif was previously described (*Opazo et al., 2008*; *Dinter et al., 2016*). WT and A53T-$\alpha$-synuclein tagged by the C-terminal and N-terminal half of Venus was obtained from Prof. Tiago Outeiro (University of Goettingen, Germany).

## Immunoblots

Immunoblots were carried out 24 h after transfection as previously described (*Dinter et al., 2016*) using NP40 lysis buffer containing protease inhibitors (Pierce, Thermo Fisher Scientific) and the following primary antibodies: rabbit anti-$\alpha$-synuclein (1:500, No. 2642, Cell Signalling Technology, Danvers, USA), mouse anti-beta-tubulin (1:1000, E7, Developmental Studies Hybridoma Bank, Iowa, USA). Secondary antibodies were anti-mouse IgG (NXA931) and anti-rabbit IgG (NA934V) from GE Healthcare Life Sciences (1:10000). These antibodies produce several nonspecific bands that are also visible in cells not expressing $\alpha$-synuclein. Among the bands around 20 kDa observed with the $\alpha$-synuclein antibody, only the upper band is considered specific and was used for quantification (see *Dinter et al., 2016* for details).

## Flow cytometry

Cells were grown in six-well plates and used 24 h after transfection. Adherent cells were washed with phosphate buffer saline (PBS) three times and detached with trypsin. Subsequently, cells were collected in FACS tubes, centrifuged for 5 min at 2000 rpm and washed again with PBS. Cell pellets were finally resuspended in 200 µl of PBS. Flow cytometry was carried out by a FACSCalibur (BD Biosciences) using forward and sideward scatter to gate cells and a fluorescence threshold of 300 AFU to detect cells with Venus (YFP) fluorescence. This threshold was determined from measurements with untransfected cells and cells expressing either the N-terminal or the C-terminal half of Venus only.

## Microscopy

For classification of EGFP distribution patterns, cells were grown on coverslips and fixed 24 h after transfection. The distribution of EGFP fluorescence was classified manually by a blinded observer into the categories 'homogenous distribution', 'containing particles' and 'unhealthy' (round, condensed cells) using an Olympus IX81 fluorescence microscope (60x oil objective, NA 1.35). At least 100 cells per coverslip were classified. In each experiment, three coverslips were evaluated per group and the results averaged.

## Drosophila stocks

Flies expressing A53T-$\alpha$-synuclein in neurons, $w[*]; ; P\{w[+mC] = GAL4 - elav.L\}$, $P\{w[+mC] = UAS - HsapSNCA.A53T\}$ and flies expressing GFP under control of GAL4 $w[*]; P(acman)\{w[+] = UAS - GFP\}5$ were previously described (*Dinter et al., 2016*). Flies expressing AS69 under control of GAL4, $w[118]; ; P\{w[+] = UAS - AS69\}$, were generated using standard P-element transformation (BestGene Inc). Expression of A53T-$\alpha$-synuclein fused to VN and VC in neurons was achieved by genetically crossing and recombining flies carrying GAL4 under the elav promoter and VN and VC tagged A53T-$\alpha$-synuclein under the UAS promoter. The resulting genotype of these flies is $P\{w[+mW.hs] = GawB\}elav[C155]; P\{w[+] = UAS - Hsap\ SNCA[A53T]: VC\}$, $PBac\{attB[+mC] = UAS - VN: Hsap\ SNCA[A53T]\}/Cyo$. Flies expressing 'always early RNAi', w[1118];

$P\{GD4261\}v13673$, were used as control in experiments conducted with the A53T-$\alpha$-synuclein VN/VC expressing flies. These flies have been shown to have no effect in genetic screens for modifiers in neurodegenerative disease models. Flies were raised and maintained at 25°C under a 12 h dark/light cycle.

## Climbing assay and fly head immunoblot

Virgins of the stock $w[*]; ; P\{w[+mC] = GAL4 - elav.L\}$, $P\{w[+mC] = UAS - Hsap\ SNCA.A53T\}$ were either crossed to males $w[118]; ; P\{w[+] = UAS - AS69\}$, or $w[*]; P(acman)]\{w[+] = UAS - GFP\}5$ (control). In the F1-progeny we selected for males with pan neural [A53T]$\alpha$-synuclein and either AS69 or GFP concomitant expression. Climbing analysis was performed 5, 15 and 25 days post eclosion as previously described (*Dinter et al., 2016*). For each time point and per genotype 10 flies were analyzed in 10 tapping experiments with 60 s resting interval and the results averaged. The crosses were repeated n = 3 times.

In parallel, 10 fly heads from the F1-progeny and also from male w[*]; P(acman)w[+]=UAS GFP flies were homogenized in 100 µl RIPA buffer using the Speedmil P12 (Analytik Jena AG). The lysates were centrifuged at 12000 rpm for 10 min and the supernatant collected and used for immunoblot analysis. The following primary antibodies were used: mouse anti-$\alpha$-synuclein (1:500, syn204, ab3309, Abcam) and mouse anti-syntaxin (1:500, 8C3, Developmental Studies Hybridoma Bank, Iowa, USA). Secondary antibody was anti-mouse IgG (NXA931) from GE Healthcare Life Sciences (1:5000).

## Fly head filter trap assay

Virgins of the stock $P\{w[+mW.hs] = GawB\}elav[C155]$, $PBac\{attB[+mC] = UAS - VN : Hsap\ SNCA[A53T]\}/Cyo$ were either crossed to $w[118]; ; P\{w[+] = UAS - AS69\}$ or $w[1118]; P\{GD4261\}v13673$ (control) males. In the F1-progeny we selected for males with pan neural [A53T]$\alpha$-synuclein and either AS69 or 'always early RNAi' concomitant expression. 10 fly heads were homogenized in 100 µl RIPA buffer using the Speedmill P12. The lysates were centrifuged at 12000 rpm for 10 min at 4°C and the supernatant collected. For the filter trap assay, equal protein amounts of RIPA fly head lysates (30 µg) were adjusted to equal volumes. An equal volume of urea buffer (8 M) was subsequently added, samples were incubated rolling at 4°C for 1 h and sonicated in a water bath for 10 min. SDS and DTT were added to a final concentration of 2% and 50 mM. Using a dot blot filtration unit, the resulting solutions were filtered through a 0.2 µm nitrocellulose membrane (Whatman) previously equilibrated with 0.1% SDS in TBS and afterwards washed in TBS-T. Membranes were further treated as an immunoblot described previously.

## Acknowledgements

AKB thanks the Leverhulme Trust and the Parkinson's and Movement Disorder Foundation (PMDF) for funding and the Novo Nordisk Foundation for support through a Novo Nordisk Foundation professorship. PF thanks the Böhringer Ingelheim Fonds and the Studienstiftung des Deutschen Volkes for support. CG thanks the Alexander von Humboldt Foundation and the European Commission (H2020-MSCA grant agreement No 706551) for support. This project has received funding from the European Research Council under the European Union's Horizon 2020 research and innovation program, grant agreement No. 726368 (to WH). We thank Nadine Rösener for help with the density gradient centrifugation and Sabine Hamm for excellent technical assistance.

## Additional information

### Funding

| Funder | Grant reference number | Author |
| --- | --- | --- |
| Leverhulme Trust | | Alexander K Buell |
| Boehringer Ingelheim Fonds | | Patrick Flagmeier |
| Studienstiftung des Deutschen Volkes | | Patrick Flagmeier |

| | | | |
|---|---|---|---|
| Alexander von Humboldt-Stiftung | | | Céline Galvagnion |
| H2020 European Research Council | 726368 | | Wolfgang Hoyer |
| Parkinson's and Movement Disorder Foundation | | | Alexander K Buell |
| Novo Nordisk Foundation | | | Alexander K Buell |
| H2020 Marie Skłodowska-Curie Actions | 706551 | | Céline Galvagnion |
| RWTH Aachen University | | START-Program of the Faculty of Medicine | Theodora Saridaki |

The funders had no role in study design, data collection and interpretation, or the decision to submit the work for publication.

## Author contributions
Emil Dandanell Agerschou, Software, Formal analysis, Investigation, Visualization, Writing—original draft, Writing—review and editing; Patrick Flagmeier, Investigation, Visualization; Theodora Saridaki, Daniel Komnig, Vibha Prasad, Formal analysis, Investigation, Writing—review and editing; Céline Galvagnion, Laetitia Heid, Investigation, Writing—review and editing; Hamed Shaykhalishahi, Resources, Investigation, Methodology; Dieter Willbold, Christopher M Dobson, Funding acquisition, Writing—review and editing; Aaron Voigt, Formal analysis, Investigation, Visualization, Writing—review and editing; Bjoern Falkenburger, Conceptualization, Formal analysis, Supervision, Funding acquisition, Methodology, Writing—original draft, Project administration, Writing—review and editing; Wolfgang Hoyer, Conceptualization, Supervision, Funding acquisition, Methodology, Writing—original draft, Project administration, Writing—review and editing; Alexander K Buell, Conceptualization, Data curation, Formal analysis, Supervision, Funding acquisition, Validation, Investigation, Visualization, Methodology, Writing—original draft, Project administration, Writing—review and editing

## Author ORCIDs
Patrick Flagmeier (iD) https://orcid.org/0000-0002-1204-5340
Daniel Komnig (iD) https://orcid.org/0000-0002-6312-5236
Dieter Willbold (iD) https://orcid.org/0000-0002-0065-7366
Aaron Voigt (iD) https://orcid.org/0000-0002-0428-7462
Bjoern Falkenburger (iD) https://orcid.org/0000-0002-2387-526X
Wolfgang Hoyer (iD) https://orcid.org/0000-0003-4301-5416
Alexander K Buell (iD) https://orcid.org/0000-0003-1161-3622

## Decision letter and Author response
Decision letter https://doi.org/10.7554/eLife.46112.sa1
Author response https://doi.org/10.7554/eLife.46112.sa2

# Additional files
## Supplementary files
• Transparent reporting form

## Data availability
Numerical data represented in the graphs for cell culture and fly experiments will be made publicly available on osf.io as we did for previous publications. The numerical data for the biophysical experiments will be made publicly available within the same repository on osf.io. The raw images of the gels used in the publication will be made publicly available. All data have been deposited on osf.io: https://osf.io/6n2gs/.

The following dataset was generated:

| Author(s) | Year | Dataset title | Dataset URL | Database and Identifier |
|---|---|---|---|---|
| Falkenburger BH, Buell AK, Agerschou ED | 2019 | An engineered monomer binding-protein for $\alpha$-synuclein efficiently inhibits the nucleation of amyloid fibrils | https://osf.io/6n2gs/ | Open Science Framework, 6n2gs |

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

## Appendix 1

### Analysis of strongly seeded aggregation data at neutral pH

In the case of aggregation experiments at high concentrations (µM) of pre-formed seeds under quiescent conditions, primary nucleation and fragmentation of $\alpha$-synuclein amyloid fibrils can be neglected (**Buell et al., 2014a**). The aggregation kinetics were analysed as previously reported by fitting a linear function to the early times of the kinetic traces (**Buell et al., 2014a**), with the exception that fitting was only performed after the initial decrease in fluorescence intensity, which is due to the temperature dependence of ThT fluorescence and a consequence of the thermal equilibration of the multiwell-plate prepared at room temperature. The fit was performed through five time points starting from the point of minimal fluorescence intensity (see **Appendix 1—figure 1**). The temperature-induced decrease in fluorescence intensity is superimposed to the increase in fluorescence due to fibril elongation. Therefore, using the initial growth rates likely leads to a small but systematic underestimation of the elongation rates. This fitting procedure was performed to obtain the values of $2k_+P(0)m(0)$, where $k_+$ is the fibril elongation rate constant, $m(0)$ the initial monomer concentration and $P(0)m(0)$ the initial number concentration of fibrils. For the comparison of the rates at different concentrations of AS69, we then calculate the ratios $r$:

$$r = \frac{\left(\frac{dM(t)}{dt}\right)_{AS69}\Big|_{t\approx0}}{\left(\frac{dM(t)}{dt}\right)\Big|_{t\approx0}} = \frac{k_+P(0)m(0,[AS69])}{k_+P(0)m(0)} \tag{9}$$

$r$ is the ratio of the initial gradient fitted to the kinetic trace for monomer elongating fibrils in the presence of AS69 and the initial gradient fitted to the kinetic trace for monomer elongating fibrils in the absence of AS69. $P(0)$ is the initial number concentration of fibrils, which is constant, as the same stock solution of seeds was used, and $m(0)$ is the initial monomer concentrations. For the prediction in **Figure 4** of the main manuscript, we calculated the equilibrium concentrations of unbound $\alpha$-synuclein, $m(0,[AS69]) = [m]_{\text{free}}$ as:

$$[m]_{\text{free}} = \frac{-([AS69]_{\text{tot}} + K_D - [m]_{\text{tot}}) + \sqrt{([AS69]_{\text{tot}} + K_D - [m]_{\text{tot}})^2 + 4K_D[m]_{\text{tot}}}}{2} \tag{10}$$

where the values obtained at different $[AS69]_{\text{tot}}$ were then used for $m(0,[AS69])$ in **Equation 9**. This procedure corresponds to the assumption that the only effect of the AS69 is to sequester soluble $\alpha$-synuclein.

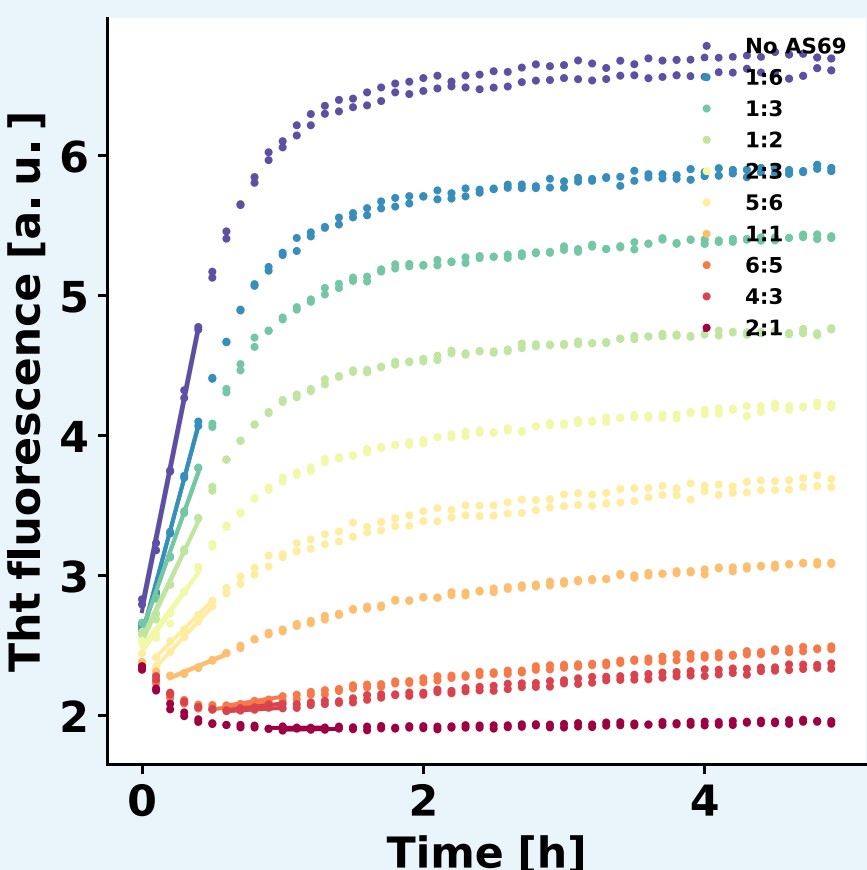

**Appendix 1—figure 1.** Linear fitting of the early times of strongly seeded aggregation kinetics. Solid lines show the fits. These data were used to produce the plot in **Figure 4c**. At the highest inhibitor concentrations, the rates were so low that the temperature increase upon introduction of the plate into the platereader led to an initial decrease in fluorescence intensity. Therefore, the data was fitted once the fluorescence intensity had started to increase.

Seeded aggregation experiments at very low monomer concentrations (0.75 µM seeds) were performed in order to test whether a concentration could be determined at which no net elongation is observed (**Appendix 1—figure 2**). The concentration of free monomer at which the rates of fibril elongation and dissociation are equal corresponds to the equilibrium concentration (**Buell et al., 2014b**):

$$k_+[m]_{\mathrm{eq}}[P] = k_-[P] \tag{11}$$

where $k_+$ is the elongation rate constant and $k_-$ is the dissociation rate constant. The equilibrium constant of monomer addition to fibril ends therefore corresponds to the inverse of the monomer concentration at equilibrium:

$$K_{\mathrm{eq}} = \frac{k_-[P]}{k_+[m]_{\mathrm{eq}}[P]} = \frac{1}{[m]_{\mathrm{eq}}} \tag{12}$$

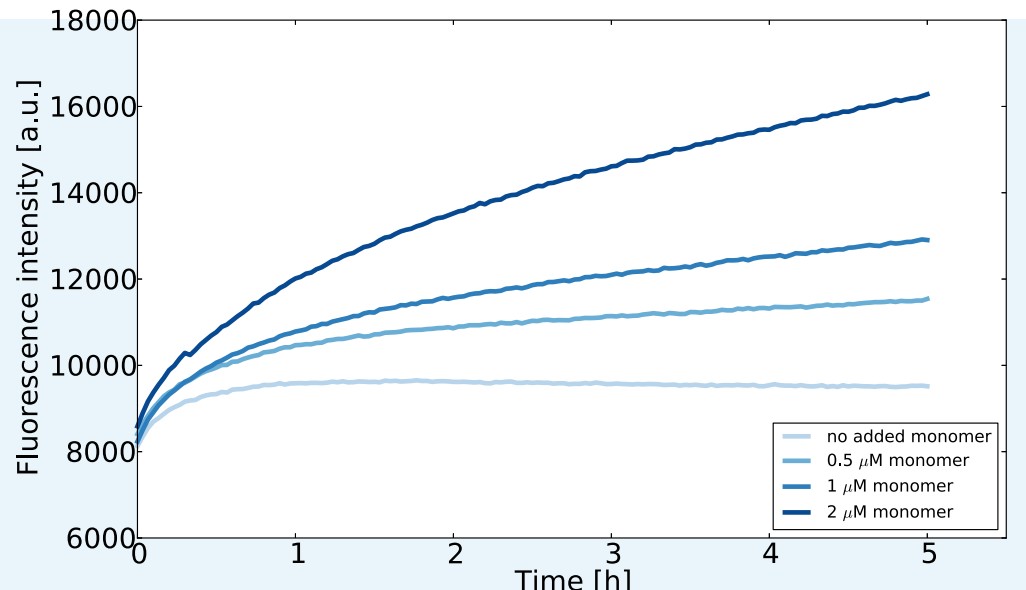

**Appendix 1—figure 2.** Seeded aggregation experiments at low monomer concentrations designed to estimate the concentration of monomeric $\alpha$-synuclein in equilibrium with fibrils. The seed concentration is in all cases 0.75 µM and the ThT concentration is 10 µM. The experiment was performed at room temperature in order to slow the reaction down and avoid temperature effects on the fluorescence upon introduction of the multiwell plate into the fluorescence platereader.

The results of these experiments are shown in *Appendix 1—figure 2*. We find that even at a concentration as low as 0.5 µM, the slight increase over time of Thioflavin-T fluorescence suggests that the fibril mass increases. This result is significant, given that the ThT fluorescence in a sample that contains only fibrils decreases over time. The fact that all samples, including that measured in the absence of added $\alpha$-synuclein monomer, show an increase in ThT fluorescence during the first hour could be explained through sedimentation processes. We have shown previously that the sedimentation of fibrils can lead to an increase in detected ThT signal if the fluorescence is read from the bottom of the multiwell plate (*Buell et al., 2014a*). However, the subsequent increase in fluorescence intensity over several hours at concentrations of 0.5 $\mu$M or higher suggests an increase in fibril mass, and hence that the critical concentration under these conditions is lower than 0.5 $\mu$M.

## Appendix 2

### Analysis of weakly seeded aggregation data at mildly acidic pH

Aggregation experiments were also performed at very low (nM) seed concentrations at mildly acidic pH and under quiescent conditions, where it has been shown that autocatalytic secondary nucleation of $\alpha$-synuclein amyloid fibrils plays an important role (**Buell et al., 2014a**). In the present study, we performed these aggregation experiments in 20 mM sodium acetate buffer at pH 5.0, well below the threshold for secondary nucleation (**Buell et al., 2014a**).

In order to quantitatively analyse the effects that AS69 and AS69fusASN exert on secondary nucleation, we started with the following equation describing the maximum aggregation rate in the presence of autocatalytic secondary nucleation (**Cohen et al., 2011**):

$$r_{\max} = \frac{M(\infty)\kappa}{e} \qquad \kappa = \sqrt{2m(0)^{n_2}[m(0)k_+ - k_{\text{off}}]k_2} \qquad (13)$$

Where $M(\infty)$ is the long time limit of the fibrillar mass concentration, $m(0)$ is the starting concentration of monomeric $\alpha$-synuclein, $n_2$ is the effective nucleus size of secondary nucleation, $k_+$ and $k_{\text{off}}$ are the rate constants of elongation and de-polymerisation respectively, and $k_2$ is the rate constant of secondary nucleation. For our analysis, we assumed the rate of de-polymerisation to be negligible and that $M(\infty)$ was not altered by the presence of AS69. Furthermore we use the upper limit of how much monomer the AS69 could possibly sequester, which is equal to the AS69 concentration. Under these assumptions, the maximum rates relative to the case where no inhibitor was present can be described as:

$$\frac{r_{\max,I}}{r_{\max,0}} = \left(1 - \frac{I}{m(0)}\right)^{\frac{n_2+1}{2}} \qquad (14)$$

Where $r_{\max,0}$ is the maximal aggregation rate in the absence of inhibitor, $r_{\max,I}$ is the maximal aggregation rate at inhibitor concentration $I$. The values of $r_{\max,I}$ for each kinetic trace were found by applying the gradient function from numpy and smoothing the resulting curves using a ten-point sliding average. The maxima of the resulting curves were taken to be $r_{\max,I}$. For the simulations, $n_2$ was varied in order to test whether the sequestration of monomer in conjunction with a higher reaction order of secondary nucleation can explain the observed strong inhibitory effect. However, even a value of $n_2$ as high as five was not able to explain the strong decrease in aggregation rate as a function of increasing inhibitor concentration. Therefore, we conclude that monomer sequestration cannot explain the highly efficient inhibition of secondary nucleation by AS69.

In the main manuscript, we discuss that the efficient inhibition of secondary nucleation by AS69 is likely to stem either from an interaction of AS69 alone or of the AS69:$\alpha$-synuclein complex with an oligomeric aggregation intermediate. Given the low population of nuclei/oligomers compared to monomers during the aggregation time course, as well as the high affinity of the AS69 for monomeric $\alpha$-synuclein, its affinity for such intermediate species would have to be significantly higher than that to monomers. This can be illustrated with a simple argument. At the end of an aggregation experiment, the fibrils typically are up to several micrometers in length, corresponding to thousands of protein molecules per fibril. Therefore, the total number of 'on pathway' oligomers that has formed during the aggregation process is three to four orders of magnitude smaller than the initial monomer concentration. In order to trap a significant fraction of these intermediates in the presence of a large excess of monomer, the affinity of AS69 to these intermediates would therefore have to be at least three orders of magnitude higher than that for monomer and hence be in the picomolar regime.

The alternative explanation, the binding of the AS69:$\alpha$-synuclein complex to the aggregation intermediate, is more plausible. A clear inhibitory effect is still observed at a ratio

$\alpha$-synuclein:AS69 of 100:1, which according to the estimate above corresponds to at least one order of magnitude more AS69:$\alpha$-synuclein complex than 'on pathway'-intermediate, rendering an efficient interference with the nucleation process plausible. Therefore, we propose a model whereby rather than requiring the binding of free AS69 to an aggregation intermediate, the AS69:$\alpha$-synuclein complex is able to incorporate into a fibril precursor and efficiently prevent it from undergoing the structural rearrangement required to transform into a growth-competent amyloid fibril.

## Appendix 3

### Determination of the lipid-induced aggregation rate

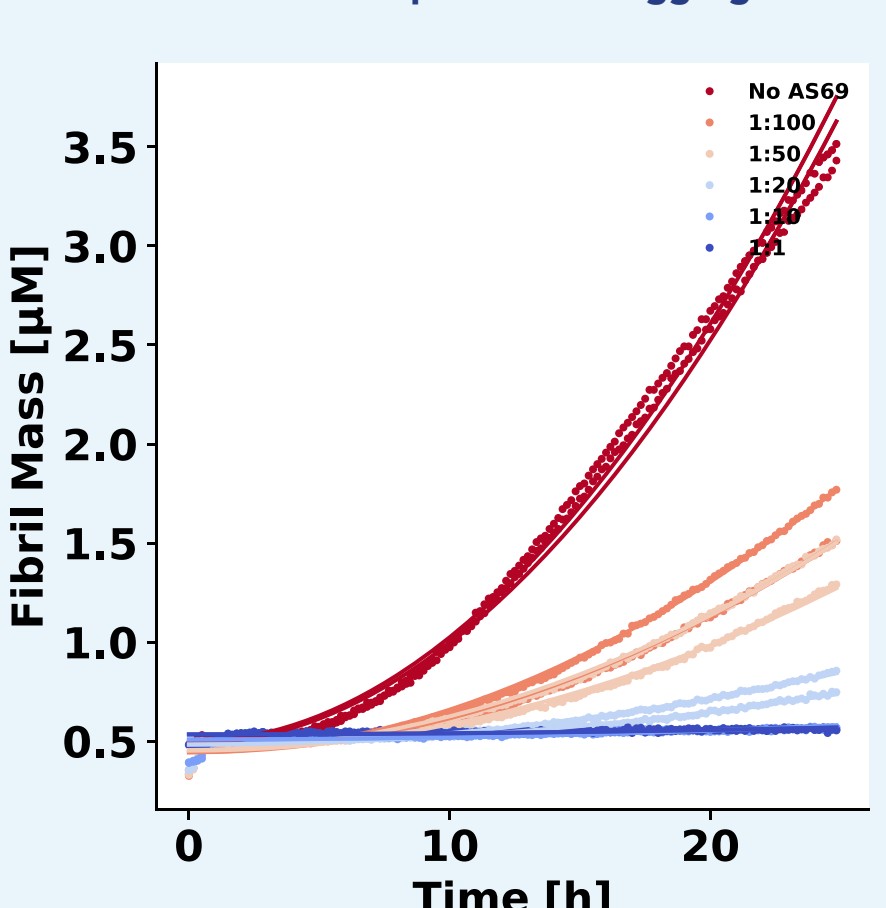

**Appendix 3—figure 1.** Analysis of the inhibition of the lipid-induced aggregation of α-synuclein by AS69.

The change in mass concentration of fibrils with time M(t) during the early time points of the lipid-induced aggregation of $\alpha$-synuclein aggregation was fitted using the single-step nucleation model described previously (**Galvagnion et al., 2015**) and the following equation (**Meisl et al., 2016**):

$$M(t) = \frac{K_M k_+ m(0)^{n+1} k_n b t^2}{2(K_M + m(0))} \qquad (15)$$

where $k_+$ is the elongation rate constant of fibrils from lipid vesicles, $k_n$ is the heterogeneous primary nucleation rate constant, $n$ is the reaction order of the heterogeneous primary nucleation reaction relative to the free monomer $m$, $b$ is the total mass concentration of the protein bound to the lipid at 100% coverage ($b = \frac{[\text{DMPS}]}{L}$, with $L$ the stoichiometry) and $K_M$ is the Michaelis constant which defines the concentration of soluble protein above which the elongation rate no longer increases linearly (fixed at 125 $\mu$M; **Galvagnion et al., 2015**). The data was normalised such that the final amount of fibril mass was set to $2b$ for the traces where no AS69 was present as it was previously shown that the fibril mass is proportional to the concentration of DMPS (**Galvagnion et al., 2015**). A quadratic equation of the form $M(t) = at^2$, was fitted to the early time points of the normalised aggregation data (see Appendix 3 subsection 3) where $a = \frac{(K_M k_+ k_n)_{\text{AS69}} b m(0, [AS69])^{n_{\text{AS69}}+1}}{2(K_{M\,\text{AS69}} + m(0, [As69]))}$. The aggregation rate, $\frac{dM(t)}{dt}$ in the presence of

AS69 normalised by the rate in the absence of AS69, for the same initial concentrations of free monomer and monomer bound to the lipid, can be computed according to:

$$r = \frac{\left(\frac{dM(t)}{dt}\right)_{AS69}}{\left(\frac{dM(t)}{dt}\right)} = \left(\frac{(K_M k_n k_+)_{AS69}\, m(0, [AS69])^{n_{AS69}+1}}{K_{M,AS69} + m(0, [AS69])}\right) \times \left(\frac{K_M + m(0)}{K_M k_n k_+ m(0)^{n+1}}\right) \tag{16}$$

In order to test whether the lipid vesicle induced aggregation of $\alpha$-synuclein in the presence of AS69 can be explained by monomer sequestration alone, we simulated the ratio $r$ for different concentrations of AS69. Starting from **Equation 15** and assuming values of $k_n k_+$, $K_M$, $b$ and $n$ independent of AS69 (which amounts to the assumption that the presence of AS69 does not change the mechanism of aggregation, but merely inhibits through depleting the free monomer) and using $n + 1 = 1.2$ (see **Galvagnion et al., 2015** for justification of $n = 0.2$) it can be shown that $r$ takes the form:

$$r = \left(\frac{m(0, [AS69])^{1.2}}{m(0)^{1.2}}\right) \times \left(\frac{K_M + m(0)}{K_M + m(0, [AS69])}\right) \tag{17}$$

Where $m(0, [AS69])$ was calculated using **Equation 10**. The result of this simulation is shown in **Figure 9c**.

