## [Decision Letter]

Thank you for submitting your article "An engineered monomer binding-protein for α-synuclein efficiently inhibits the proliferation of amyloid fibrils" for consideration by *eLife*. Your article has been reviewed by three peer reviewers, including Andrew B West as the Reviewing Editor and Reviewer #1, and the evaluation has been overseen by Huda Zoghbi as the Senior Editor. The following individuals involved in review of your submission have agreed to reveal their identity: Laura A Volpicelli-Daley (Reviewer #2).

The reviewers have discussed the reviews with one another and the Reviewing Editor has drafted this decision to help you prepare a revised submission.

The work by Buell et al., follows on from earlier work that further characterizes a binding partner of α-synuclein that can mitigate toxic conversion, especially to fibrillar assemblies. The authors show in vitro and in simple models that AS69 inhibits fibril elongation and fibril secondary nucleation. Interestingly the inhibition of secondary nucleation by AS69 can occur in the presence of excess monomer. These experiments may well extend to other proteins with amyloidogenic properties linked to disease.

Essential revisions:

1) The BiFC assays are over-interpreted and not convincing. α-synuclein can form native oligomers that are normal and functional (Dettmer et al., 2015), or oligomers that are β-sheet and toxic/pathogenic (Chen et al., 2003). Using the BiFC assay, can a PMCA assay using lysates from these cells be performed to show that the aggregates formed in the HEK293 can seed fibril formation? Are the aggregates insoluble in nonionic detergents? Do the aggregates produce 10-15 nm filaments in electron microscopy as shown in Parkinson's disease brains? Alternatively, more advanced HEK-fibril-seeding cell-based assays (also involving HEK-cells over-expressing α-synuclein) are now widely in use that provide more specific endpoints and quantifiable features specific to fibril formation. Some reviewers feel it is imperative to study α-synuclein oligomerization without the use of tags that alter the kinetics and binding.

2) For the fly model, it is not clear that this experiment has all the necessary controls. What about flies with no GFP or AS69? Or flies with AS69+GFP? The dot blot of "aggregated synuclein" is not sufficient: common in analysis is sequential extraction of α-synuclein extraction in RIPA, followed by extraction of the pellet in 2% SDS, then extraction of any remaining pellet in Urea. Does AS69 prevent α-synuclein from forming insoluble aggregates? (see Davis et al., 2016, Figure 4 and Figure 5 as examples). For these data, it is important to show the entire blot instead of blots cropped at 15 kDa to determine if higher molecular weight synuclein is formed. Do the flies show loss of dopamine neurons that are rescued by AS69?

3) A critical missing piece from the in vitro assay set is the effect of substrate binding on spontaneous fibril formation, which is different from low-concentration seeding assays. Acknowledging increased variability associated with spontaneous fibril formation, assays could still be devised (and have been done in past studies) in normalized way that would lead to new conclusions that would either support or refute the present hypothesis from seeded-based in vitro assays. Further, AS69 should be demonstrated to interact with monomer and not fibril in a density gradient run. Right now, the monomer control interacting with AS69 under centrifugation conditions is missing.

4) Further information is needed on AS69 purification, all blots should be presented in full form, and full methods in place. There are no space limitations that should affect the way data are presented, no reason to significantly cross-reference other studies for methods and tools, and experiments need to be completed with full controls.

---

## [Author Response]

Essential revisions:1) The BiFC assays are over-interpreted and not convincing. α-synuclein can form native oligomers that are normal and functional (Dettmer et al., 2015), or oligomers that are β-sheet and toxic/pathogenic (Chen et al., 2003).

We appreciate that α-synuclein forms various oligomeric species that display very different degrees of cytotoxicity, ranging from physiological oligomers to highly cytotoxic oligomers. We also acknowledge that addition of any label, in particular the two halves of the protein Venus, modifies the free energy landscape of α-synuclein oligomerisation. Still, in spite of these shortcomings, the BiFC assay is to our knowledge the most widely used readout for self-interaction of α-synuclein in living cells. We used the BiFC assay to obtain the statistically very robust finding that AS69 is indeed able to interfere with this self-interaction inside living cells.

In response to the reviewers’ concerns, we have removed the notion that AS69 reduced the occurrence of α-synuclein oligomers from the Abstract, from the subsection “Co-expression of AS69 reduces visible α-synuclein aggregates in cell culture”. The remaining interpretation of the BiFC now reads “This finding is consistent with the hypothesis that the effects of AS69 in this cellular model system result from the inhibition of a direct interaction between α-synuclein molecules, and not from an enhanced clearance of α-synuclein.” This interpretation may sound trivial given all that we know about the effects of AS69 on α-synuclein in vitro. Yet, the complex environment in living cells easily might have affected the integrity of AS69 or its affinity for α-synuclein.

We note that the purpose of the cell culture experiments was to determine whether α-synuclein behaves qualitatively similarly in living cells as in a test tube. Therefore, we used tagged α-synuclein constructs in order to report different aspects of α-synuclein behavior in living cells. We are aware that cultured HEK293 cells are not a model of Parkinson’s disease in the strict sense. In order to demonstrate that the effects of AS69 are functionally relevant in neurons and in an intact organism we used the fly experiments where the detrimental effects of untagged α-synuclein on fly motor function were rescued by AS69.

Using the BiFC assay, can a PMCA assay using lysates from these cells be performed to show that the aggregates formed in the HEK293 can seed fibril formation?

We have attempted, in the short time available for the revision, to establish such an assay and we attach the corresponding data as a separate file. We have tested the seeding efficiency of extracts of the two different cell models (tagging with halves of Venus as used for BiFC and flexible tagging with EGFP as used for counting of visible aggregates), as well as lysates of fly heads, and we found no clear seeding activity in any of these systems. Yet, based on a detailed characterization of the amount of seeds detected by this assay, we found the assay rather insensitive. Therefore, we cannot exclude that the assemblies formed in these biological models can act as seeds. We have decided not to include this negative data into the manuscript but are prepared to include them if the reviewers consider them a useful addition to the paper.

Are the aggregates insoluble in nonionic detergents? Do the aggregates produce 10-15 nm filaments in electron microscopy as shown in Parkinson's disease brains?

With the flexible EGFP tagging approach we observed visible aggregates that stained with a fibril-specific antibody provided by Dr. CG Glabe (Opazo et al., 2008) and amorphous protein aggregates in cells using electron microscopy (Saridaki et al., 2018). With the BiFC constructs, we did not observe significant numbers of visible aggregates using fluorescence microscopy. In order to detect α-synuclein species with seeding capacity, we carried out the RT-QuIC assay described in the attached file.

Since the ultimate aim of our studies is the development and characterisation of potential therapeutic strategies, quantitative assays are important. We have not established electron microscopy of cell culture lysates because the amount of 10-15 nm filaments appeared hard to quantify reliably.

Alternatively, more advanced HEK-fibril-seeding cell-based assays (also involving HEK-cells over-expressing α-synuclein) are now widely in use that provide more specific endpoints and quantifiable features specific to fibril formation.

This is an excellent suggestion, but performing such an assay would have required significantly more time than the period of revision, given that it is not established in our laboratories. We agree that this should be done in the future, probably in collaboration with a laboratory where this type of assay is established. We therefore have added a sentence at the end of the manuscript (second last sentence), which reads: “Further steps will be to test the effects of AS69 in cell-based fibril seeding assays, in mammalian dopaminergic neurons, and in PD models where aggregates are formed from endogenous α-synuclein.”

Some reviewers feel it is imperative to study α-synuclein oligomerization without the use of tags that alter the kinetics and binding.

We agree that from a basic physico-chemical viewpoint, the addition of a label is a major change to the system that is likely to influence its behaviour. In particular in the cell-based models, fluorescent labelling is a pre-requisite in order to be able to analyse a sufficiently large number of cells to obtain robust statistics. However, in order to determine the effect of AS69 on untagged α-synuclein we used neuronal expression of untagged A53T-α-synuclein for the fly experiments depicted in Figure 3A,B. In the original text, the fact that untagged α-synuclein was used for Figure 3A,B was not stressed enough. We therefore added “untagged” in subsection “Co-expression of AS69 rescues A53T α-synuclein dependent phenotype in *Drosophila melanogaster*”.

Overall, we believe that the combination of our highly detailed biophysical in vitro experiments with unlabelled protein with the robust readouts obtained with both labelled and unlabelled protein in vivo represents a powerful strategy for the characterisation of the potential therapeutic benefits of AS69, as well as their mechanism.

2) For the fly model, it is not clear that this experiment has all the necessary controls. What about flies with no GFP or AS69? Or flies with AS69+GFP?

The α-synuclein phenotype is concentration-dependent, both in humans and in flies. Comparing flies where GAL4 drives only α-synuclein to flies where UAS drives both α-synuclein and AS69 would be inappropriate because more α-synuclein would be produced in the first case than in the second. Therefore, to “dilute” the GAL4 to a similar event in both lines, we compared UAS driving α-synuclein and AS69 flies to UAS driving α-synuclein and GFP. We are not sure that the control AS69+GFP would add any information in this context.

The dot blot of "aggregated synuclein" is not sufficient: common in analysis is sequential extraction of α-synuclein extraction in RIPA, followed by extraction of the pellet in 2% SDS, then extraction of any remaining pellet in Urea. Does AS69 prevent α-synuclein from forming insoluble aggregates? (see Davis et al., 2016, Figures 4 and 5 as examples). For these data, it is important to show the entire blot instead of blots cropped at 15 kDa to determine if higher molecular weight synuclein is formed.

We think there must be a small misunderstanding. The filter trap assay of the fly head lysates is not cropped at 15 kDa, the entire blot is already shown. The blot cropped at 15 kDa is the western blot of HEK293 cell lysates. We show only the 15 kDa region in the figure because the antibody displays many strong nonspecific bands at higher molecular weight that obscure the interesting changes. We have included the entire western blot as Figure2—figure supplement 2 and describe specific and nonspecific bands.

As for the filter trap assay, we agree that there are established protocols to determine various degrees of detergent and denaturant solubility of α-synuclein aggregates in flies. Yet, not all research groups use the same extraction protocol. We have not obtained reliable results with this method and therefore do not use it routinely in our studies (see e.g. our recent publication on α-synuclein multimerization in flies: Prasad et al., 2019). Here, we use the filter trap assay because it provides a measure of large α-synuclein aggregates (i.e. large enough to be retained in the filter). We concede, however, that this is not a statement about the structural and biophysical nature of these aggregates.

Do the flies show loss of dopamine neurons that are rescued by AS69?

Flies harbor only few dopaminergic neurons. This makes it hard to quantify rescue of α-synuclein-dependent degeneration. In addition, dopaminergic neurons in flies are very different from dopaminergic neurons in mammals. We consider mice the more appropriate model system to study α-synuclein effects on dopaminergic neurons. Experiments with AS69 and α-synuclein in the nigrostriatal pathway of mice are currently underway. We have added a sentence at the end of the manuscript (second last sentence), which reads: “Further steps will be to test the effects of AS69 in cell-based fibril seeding assays, in mammalian dopaminergic neurons, and in PD models where aggregates are formed from endogenous α-synuclein.”

3) A critical missing piece from the in vitro assay set is the effect of substrate binding on spontaneous fibril formation, which is different from low-concentration seeding assays. Acknowledging increased variability associated with spontaneous fibril formation, assays could still be devised (and have been done in past studies) in normalized way that would lead to new conclusions that would either support or refute the present hypothesis from seeded-based in vitro assays.

We have put great efforts in recent years into the establishment of a well-defined and reproducible set of conditions to probe the spontaneous formation of α-synuclein amyloid fibrils. The only paradigm where we have been able to perform a highly quantitative kinetic analysis is the lipid (SUVs of DMPS or other negatively charged lipids)-induced aggregation of α-synuclein. In this context, it is important to note that the most widely employed conditions for de novo α-synuclein aggregation (strong agitation/shaking, binding surfaces) have not yet been possible to subject to a quantitative kinetic analysis. We have therefore opted to employ also lipid-induced aggregation in the present study and have performed extensive additional kinetic (ThT fluorescence) and thermodynamic (CD spectroscopy, calorimetry) experiments, which are all included in the revised form of the manuscript. We find a highly efficient inhibition as well and we show that monomer sequestration is unable to account for this effect, suggesting that a similar mechanism of inhibition as for secondary nucleation might be at play. However, complementary calorimetric experiments show that the AS69 is able to directly interact with the lipid vesicles, possibly leading to an additional inhibitory effect (see Brown et al., 2016). We are at present unable to define the contributions of each of these two (in addition to monomer sequestration) mechanisms of inhibition to the overall inhibitory effect. Nevertheless, these experiments demonstrate that AS69 is very effective even under conditions where the contribution of secondary nucleation is negligible.

Further, AS69 should be demonstrated to interact with monomer and not fibril in a density gradient run. Right now, the monomer control interacting with AS69 under centrifugation conditions is missing.

We thank the reviewers for this suggestion, and we have added structural and calorimetric data, summarized in the new Figure 1, that unambiguously demonstrate that α-synuclein and AS69 bind with a 1:1 stoichiometry and high affinity at pH values ranging from 7.4 to 5. We think that this data is even stronger than potential density gradient centrifugation experiments in characterizing this interaction.

4) Further information is needed on AS69 purification, all blots should be presented in full form, and full methods in place. There are no space limitations that should affect the way data are presented, no reason to significantly cross-reference other studies for methods and tools, and experiments need to be completed with full controls.

We have added a detailed description of the protocol for the production of AS69, and we have added the raw data of flow cytometry assays and full blots (see other points).